# Optimal Data Sampling for Training Neural Surrogates of Programs

## Abstract

Programmers and researchers are increasingly developing *surrogates* of programs, models of a subset of the observable behavior of a given program, to solve a variety of software development challenges. Programmers train surrogates from measurements of the behavior of a program on a dataset of input examples. We present a methodology for optimally sampling datasets to train neural network based surrogates of programs. We first characterize the optimal proportion of data to sample from each path in a program based on the complexity of learning the path. We next provide a program analysis to determine the complexity of different paths in a program. We evaluate these results on a large-scale graphics program, demonstrating that theoretically optimal sampling results in empirical improvements in accuracy.

## 1 Introduction

Programmers and researchers are increasingly developing *surrogates* of programs, models of a subset of the observable behavior of a given program, to solve a variety of software development challenges (Renda et al., 2021). For example, Esmaeilzadeh et al. (2012) train small neural networks to mimic existing programs, then deploy the neural networks in place of the programs to speed up computation. Generally, surrogates are used to accelerate programs (Esmaeilzadeh et al., 2012; Mendis et al., 2019; Munk et al., 2019), apply transfer learning to programs (Tercan et al., 2018; Kustowski et al., 2020; Kwon & Carloni, 2020), and approximate the gradient of programs to optimize their inputs (Renda et al., 2020; She et al., 2019; Tseng et al., 2019).

**Dataset generation.** Training a surrogate of a program requires measurements of the behavior of the program on a dataset of input examples. There are three common approaches to collecting this dataset. The first is to use data that is uniformly sampled (or sampled using another manually defined distribution) from the input space of the program (Tseng et al., 2019; Kustowski et al., 2020). The second is to use data instrumented from running the original program on a workload of interest (Renda et al., 2020; Esmaeilzadeh et al., 2012). The third is to use *active learning* (Settles, 2009), a class of online methods that iteratively query labels for data points based on the expected improvement in accuracy resulting from additional samples (İpek et al., 2006; She et al., 2019; Pestourie et al., 2020).

These approaches show promise, but they face challenges with programs with control flow. Programs with control flow (e.g. branches and loops) are piecewise functions: each control flow *path* induces a different *trace* of operations that are applied to the input. The sampling techniques above do not optimally allocate samples between different paths, resulting in surrogates which do not adequately learn the behavior of the program along all paths. For example, Renda et al. (2020, Section IV.A) identify a scenario in which an instrumented dataset does not exercise a set of paths in the program enough times for the surrogate to learn the behavior along those paths.

**Our approach.** Our approach uses the source code and semantics of the program under study to guide dataset generation for training a surrogate of the program. The core concept is to analyze the complexity of each path in a program and to allocate more samples to paths that are more complex to learn.

**Stratified functions.** Our approach represents the program as a *stratified function*, a function with different behavior in different regions (*strata*) of the input space (i.e., a piecewise function).[1] We use

---

[1] We choose the term *stratified* by analogy with the technique of stratified sampling.

*stratified surrogates* to model such functions. To construct a stratified surrogate, we train independent surrogates of each component of the stratified function. At evaluation time, a stratified surrogate checks which stratum an input is in (using the original program) then applies the corresponding surrogate.

This evaluation-time stratum check must not preclude the use of the surrogate for its downstream task. We therefore adopt a standard modeling assumption in the approximate computing literature: that precisely determining paths is an acceptable cost during approximate program execution (Sampson et al., 2011; Carbin et al., 2013).[2,3]

**Optimal sampling.**     With this stratified modeling assumption, we then determine how many samples to allocate to train each surrogate. Using neural network sample complexity bounds for learning analytic functions (Arora et al., 2019; Agarwala et al., 2021) we calculate a *complexity* for each component function which gives an upper bound on how many samples are required to learn the behavior of that component to a given error. Given a data distribution describing the frequency of each component and given each component function's complexity, we then derive the optimal number of samples to allocate to training each surrogate of each component, minimizing the upper bound on the stratified surrogate's error.

**Complexity analysis.**     We present a programming language, TURACO, in which programs denote stratified functions with well-defined complexity measures. We provide a program analysis for TURACO programs that automatically determines the strata of the function and calculates an upper bound on the complexity of each component of the stratified function that the program denotes.

**Renderer demonstration.**     To demonstrate that optimal sampling using our complexity analysis improves surrogate accuracy on downstream tasks, we present a case study of learning a surrogate of a renderer in a video game engine. We show that our optimal sampling approach results in between $15\%$ and $47\%$ lower error than training using distributions that do not take into account path complexity. These accuracy improvements correlate with perceptual improvements in the generated renders.

**Contributions.**     In sum, we present the following contributions:

- An optimal approach to allocating samples among strata to train stratified neural network surrogates of stratified analytic functions that minimizes the upper bound on the surrogate's error.
- A programming language, TURACO, in which all programs are learnable stratified functions, and a program analysis to determine the complexity of learning surrogates of those programs.
- An evaluation of these results on a graphics program, demonstrating that theoretically optimal sampling using TURACO's complexity analysis results in empirical improvements in accuracy.

We lay the groundwork for analyzing optimal sampling approaches for training surrogates of programs. Our results hold out the promise of surrogate training approaches that intelligently use the program's semantics to guide the design and training of surrogates of programs.

## 2    EXAMPLE

Figure 1a presents an example distilled from our evaluation (Section 5) that we use to demonstrate how *optimal path sampling*, sampling from paths according both to their frequency in a data distribution and to their complexity, results in a more accurate surrogate than *frequency-based path sampling*, sampling according to the frequency of paths alone.

**Program under study.**     We study a graphics program that calculates the luminance (i.e., brightness) at a point in a scene as a function of `sunPosition`, the height of the sun in the sky (i.e., the time of day) which ranges from $-1$ to $1$, and `emission`, a property of the material at that point which ranges from $-1$ to $1$. The program first checks whether it is daytime (Line 2), and sets the ambient lighting variable accordingly. The program next checks whether the sun position is above a threshold (Line 7) and sets the emission variable accordingly. The output is then the sum of the ambient light and the light emitted by the material. Figure 1b presents the output of this program on inputs between $-1$ and $1$.

---

[2]"EnerJ ... prohibit[s] approximate values in conditions that affect control flow." (Sampson et al., 2011).

[3]"Rely assumes that ... control flow branch targets are computed reliably." (Carbin et al., 2013).

```
1 fun (sunPosition, emission) {
2   if (sunPosition < 0) {
3     ambient = 0
4   } else {
5     ambient = sunPosition
6   }
7   emission *= max(0.1, sunPosition)
8 } return ambient + emission
```

Luminance Across Input Distribution

(a) Graphics program calculating the luminance of a pixel as a function of ambient light and material properties.

(b) Output of the program on inputs in $[-1, 1]$, with dashes separating the three paths.

```
// assume: sunPosition < 0
fun (sunPosition, emission) {
  ambient = 0;
  emission *= 0.1;
} return ambient + emission;
```

```
// assume 0 < sunPosition < 0.1
fun (sunPosition, emission) {
  ambient = sunPosition;
  emission *= 0.1;
} return ambient + emission;
```

```
// assume: sunPosition > 0.1
fun (sunPosition, emission) {
  ambient = sunPosition;
  emission *= sunPosition;
} return ambient + emission;
```

(c) Nighttime (ll) path.

(d) Twilight (rl) path.

(e) Daytime (rr) path.

Figure 1: Example program, outputs, and traces.

The path conditions (Lines 2 and 7) partition the program into three traces: nighttime, when sunPosition is less than 0 (Figure 1c); twilight, when sunPosition is between 0 and 0.1 (Figure 1d); and daytime, when sunPosition is greater than 0.1 (Figure 1e). These paths are separated by dashed black lines in Figure 1b.

Training a surrogate of this program poses a particular challenge because of the different behavior of these traces. Furthermore, these traces have different relative complexities: when sunPosition is less than 0.1 the function is linear, but when sunPosition is above 0.1 the function is quadratic.

We must ensure that the data distribution that we use for training surrogates reflects not only the different paths of the program, but also the relative complexities of each path of the program.

**Optimal path sampling.** We present an approach to determining the optimal amount of training data to sample from each path to train a stratified surrogate of this program. Specifically, given a data distribution and a data budget we want to find the optimal number of data points to sample from each path to minimize the expected error of a surrogate of the program over the data distribution. Intuitively, our approach is to prioritize sampling paths that are frequent and paths that are complex (and thus require more samples to learn).

First we determine the frequency of each path in the underlying data distribution. We assume that the data has a uniform distribution over inputs between $-1$ and $1$. This results in path frequencies for the nighttime path (sunPosition < 0) of $50\%$, the twilight path (0 < sunPosition < 0.1) of $10\%$, and the daytime path (0.1 < sunPosition) of $40\%$.

Next we determine the *sample complexity* of each path, the number of samples required to learn the function along that path to a given error. We use the sample complexity results of Agarwala et al. (2021), who give an upper bound on the number of samples required to learn a neural network approximation of a given function. Using this bound (as implemented by our TURACO analysis described in Section 4.2), we determine that the twilight path takes $1.5\times$ as many samples to train a surrogate to a given error as the nighttime path, and the daytime path requires $5\times$ as many samples.

Finally, we combine these metrics to determine the optimal sampling rates. Using Equation (3) in Section 3.2, we find that the optimal sampling rate is to sample $39.3\%$ of the data from the nighttime path, $14.3\%$ of the data from the twilight path, and $46.4\%$ of the data from the daytime path.

**Stratified surrogates.** The class of surrogate model for which the above approach is optimal is that of a *stratified neural surrogate* – a set of disjoint neural networks which are applied based on which

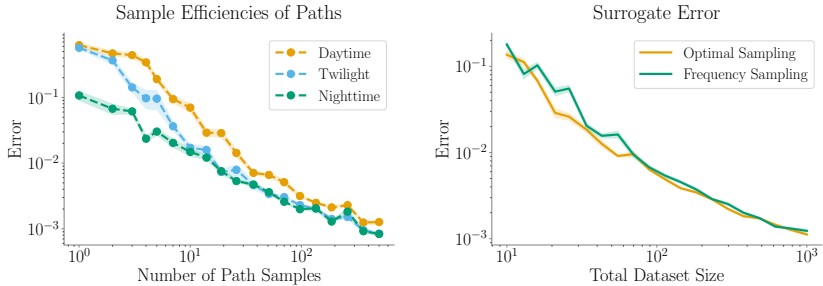

(a) Per-path surrogate errors (log-log plot).    (b) Stratified surrogate errors (log-log plot). Optimal sampling decreases the error by $15\%$.

Figure 2: Per-path surrogate errors (left) and combined errors (right) for the example.

path the inputs induce in the program. Concretely, this means that we train one surrogate per path, and pick which to apply for each input at evaluation time. For this example program, picking which surrogate to apply just requires comparing `sunPosition` against constant threshold values.

**Training methodology.** For each surrogate, we train a 1-hidden-layer MLP with 512 hidden units with a ReLU activation, using 10,000 steps of Adam with learning rate $0.0005$ and batch size 128.

**Results.** Figure 2 presents the error of surrogates trained to mimic this program, as a function of the training dataset size. On the left, Figure 2a presents the error of surrogates of each path. On the right, Figure 2b presents the error of stratified surrogates of the entire program for optimal path sampling and a baseline of sampling according to path frequency alone. The x axis of each plot is the dataset size used to train the surrogate (on the left, the dataset size per path; on the right, the total dataset size used for all paths). The y axis of each plot is the error of the resulting surrogate (lower is better).

Figure 2a shows that though the complexity measure is not exactly proportional to the empirical sample complexity, it does correlate with it: at a given data budget, the daytime path with the highest complexity has the highest error, followed by twilight then nighttime.

Figure 2b shows that the optimal path sampling approach results in lower error than sampling according to path frequency alone. For datasets of total size below 70 samples, the surrogate trained with optimal path sampling has a geometric mean decrease in error of $27.5\%$. For datasets of total size above 70 samples, the surrogate trained with optimal path sampling has a geometric mean decrease in error of $5.5\%$. Across the entire range of dataset sizes evaluated in this plot, the surrogate trained with optimal path sampling has a geometric mean decrease in error of $15\%$.

## 3   OPTIMAL SAMPLING

In this section, we formally define stratified functions and stratified surrogates, and derive the optimal sampling distribution to use when training a stratified surrogate of stratified function.

### 3.1   SETUP

We define a *learning algorithm*, a function that trains a surrogate of a given input function, as a random function $tr : (\mathcal{X} \to \mathcal{Y}) \times \mathcal{D} \times \mathbb{N} \times \mathcal{L} \to (\mathcal{X} \to \mathcal{Y})$ that takes a function $f : \mathcal{X} \to \mathcal{Y}$ from inputs $x \in \mathcal{X}$ to outputs $y \in \mathcal{Y}$, a distribution $D \in \mathcal{D}$ over inputs $x$, a number of training examples $n \in \mathbb{N}$, and a loss function $\ell \in \mathcal{L} : \mathcal{Y} \times \mathcal{Y} \to \mathbb{R}_{\geq 0}$ which measures the cost of an incorrect prediction, and returns a function (representing the output surrogate) $\hat{f} : \mathcal{X} \to \mathcal{Y}$.

Equation 1 defines a given function $f$ as *probably approximately correctly learnable* (abbreviated as learnable for the remainder of the paper) for a given learning algorithm $tr$ and loss function $\ell$ if for all distributions $D$, with high probability $1 - \delta$ the learning algorithm returns a surrogate $\hat{f}$ that

approximately matches the original function $f$ over the distribution $D$:

$$\forall D, \epsilon \in (0,1), \delta \in (0,1). \exists n. \underset{\hat{f} \sim tr(f,D,n,\ell)}{P}\left(\underset{x \sim D}{\mathbb{E}}\left[\ell\left(\hat{f}(x), f(x)\right)\right] \leq \epsilon\right) \geq 1 - \delta \tag{1}$$

Following Arora et al. (2019) and Agarwala et al. (2021) we study functions and learning algorithms for which there is a measure of the complexity $\zeta(f)$ of function $f$ for the learning algorithm $tr$ such that the relationship between $n$, $\zeta(f)$, $\epsilon$, and $\delta$ in Equation (1) is:

$$\exists C. n \leq C\left[\frac{\zeta(f) + \log\left(\delta^{-1}\right)}{\epsilon^2}\right] \tag{2}$$

We instantiate the complexity measure $\zeta(f)$ for neural networks in Section 3.3.

We define a *stratified function* $f$ as follows:

$$f(x) \triangleq \begin{cases} f_1(x) & \text{if } x \in s_1 \\ \vdots \\ f_c(x) & \text{if } x \in s_c \end{cases}$$

where $c$ is the number of strata, $\{s_i\}_{i=1}^c$ are strata, $\forall i \neq j. \, s_i \cap s_j = \varnothing$, and $\cup_i s_i = \mathcal{X}$.

We define a *stratified surrogate* $\hat{f}$ as a stratified function with components $\hat{f}_i$.

For a data distribution $D$, let $D(x)$ be the probability that $x$ is sampled from $D$, and $D(s_i)$ be the total probability mass of all data points within $s_i$ over $D$ (i.e., $D(s_i) = \int_{x \in s_i} D(x)$). Let $D(x|s_i)$ be the probability of a data point $x$ sampled from $D$ if $x \in s_i$.

## 3.2 Optimal Sampling

Our goal is to learn a stratified surrogate $\hat{f}$ of a stratified function $f$, where each component function $f_i$ is learnable. We are given a data distribution $D$, a maximum sample budget $n$, a learning algorithm $tr$, a loss function $\ell$, and a failure probability $\delta$. Our task is therefore to find the number of samples to allocate to each stratum to train a surrogate of that function component. We assume that each surrogate component's failure probability $\delta_i = \frac{\delta}{c}$, which satisfies the overall failure probability by union bound.

Note that Equation (2) is an upper bound and not an exact equality. We are therefore minimizing the upper bound of the error of the resulting surrogate, rather than directly minimizing the error.

Formally, we solve the following optimization problem:

$$\underset{\{n_i\}_{i=1}^c \in \mathbb{N}^+}{\text{argmin}} \, \underset{x \sim D}{\mathbb{E}}\left[\ell\left(\hat{f}(x), f(x)\right)\right] \text{ s.t. } \sum_{i=1}^c n_i \leq n \text{ and } \hat{f}_i \sim tr(f_i, D(x|s_i), n_i, \ell) \tag{3}$$

**Theorem 3.1.** *With $\delta_i = \frac{\delta}{c}$, the upper bound of Equation (3) is minimized at:*

$$n_i = n \frac{\left(D(s_i)\sqrt{\zeta(f_i) + \log(c\delta^{-1})}\right)^{\frac{2}{3}}}{\sum_{i=1}^c \left(D(s_i)\sqrt{\zeta(f_i) + \log(c\delta^{-1})}\right)^{\frac{2}{3}}},$$

The proof of this theorem is presented in Appendix C.

## 3.3 Neural Network Learnability

Equation (2) defines the required sample complexity for learning $\hat{f}$ as a function of $\zeta(f)$, the complexity of $f$. This section defines $\zeta(f)$ for training neural network surrogates of analytic functions. This section is an abridged summary of assumptions and results presented by Agarwala et al. (2021) and Arora et al. (2019); refer to Agarwala et al. (2021) for the full set of assumptions.

Agarwala et al. (2021) provide a calculus for learning surrogates of analytic functions $f$ (around 0) based on the *tilde* $\tilde{f}$ of the function:

$$f(x) = \sum_{n=0}^{\infty} a_n x^n \qquad\qquad \tilde{f}(x) \triangleq \sum_{n=0}^{\infty} |a_n| x^n$$

Note the following properties for $x \geq 0$:

$$\tilde{h}(x) \leq \begin{cases} \tilde{f}(x) + \tilde{g}(x) & \text{if } h(x) = f(x) + g(x) \\ \tilde{f}(x) \cdot \tilde{g}(x) & \text{if } h(x) = f(x) \cdot g(x) \\ \tilde{f}(\tilde{g}(x)) & \text{if } h(x) = f(g(x)) \text{ and } \tilde{f} \text{ converges for } \tilde{g}(x) \end{cases} \tag{4}$$

For a 2-layer neural network trained with stochastic gradient descent, if $f$ is analytic, $\vec{x}$ is on the $d$-dimensional unit sphere ($\vec{x} \in S^d$), $\beta \in \mathbb{R}^d$ (a parameter set to control the scale of the inputs), and $\ell$ is 1-Lipschitz, then $f(\beta \cdot \vec{x})$ is learnable with:

$$\zeta(f) = \left( \|\beta\|_2 \tilde{f}'(\|\beta\|_2) + \tilde{f}(0) \right)^2 \tag{5}$$

## 4 TURACO: PROGRAMS AS STRATIFIED FUNCTIONS

In this section we present TURACO, a programming language in which all programs denote learnable stratified functions. We provide a program analysis for TURACO programs which calculates an upper bound on the complexity of each component of the stratified functions that the program denotes.[4]

### 4.1 SYNTAX AND STANDARD INTERPRETATION

$$p ::= \text{fun}\,(x+)\,\{\,s\,\}\,\text{return}\,x$$
$$s ::= \text{skip} \mid s;s \mid x := e$$
$$\qquad \mid \text{if}\,(e>0)\,\{\,s\,\}\,\text{else}\,\{\,s\,\}$$
$$e ::= x \mid v \mid b(e,e) \mid u(e)$$
$$b ::= \text{ADD} \mid \text{MUL}$$
$$u ::= \text{NEG} \mid \text{SIN} \mid \text{EXP} \mid \text{LOG1P}$$
$$x ::= \text{set of variable names}$$
$$v ::= \text{set of floating point values}$$

Figure 3: Syntax of TURACO.

Figure 3 presents the syntax of TURACO, a loop-free IMP-like language (Winskel, 1993). A TURACO program $p$ takes a list of inputs $x$, executes a top-level statement $s$, and returns a single variable $x$. Statements $s$ are skips, sequences, assignments, or if statements. Expressions $e$ are variables $x$, values $v$, binary operations $b$, or unary operations $u$.

TURACO supports analytic operations (e.g., NEG, SIN, EXP), including those which can be represented by a power series within a given domain: LOG1P computes $\log(1+x)$ for $x \in (-1,1]$. We restrict the supported operations to those required to implement the case study in Section 5.

Appendix D.1 presents the full set of semantics for TURACO.

### 4.2 COMPLEXITY ANALYSIS

We now present a program analysis that gives an upper bound on the complexity of traces of TURACO programs. This section presents a core set of rules; Appendix D.2 presents the full analysis.

The analysis uses three core concepts: a *complexity interpretation* of expressions to calculate an upper bound on the tilde of expressions based on the calculus in Equation (4), a *dual-number execution* (Wengert, 1964; Griewank & Walther, 2008) to calculate the derivative of the upper bound on the tilde (to compute the first term in Equation (5)), and a *path analysis* which splits the program by paths to compute the complexity of each trace.

Figure 4 presents the big-step relations used to calculate an upper bound on the complexity for a subset of expressions. The relation $\langle \tilde{\sigma}, [e] \rangle \tilde{\Downarrow} (\tilde{v}, \tilde{v}')$ says that under the variable complexity mapping $\tilde{\sigma}$ (mapping variables to tuples with their respective tilde and tilde derivative), the expression $e$ has $\tilde{e} \leq \tilde{v}$ and $\tilde{e}' \leq \tilde{v}'$. Note that the rule for MUL uses the upper bound for multiplication in Equation (4).

Figure 5 presents the complexity relation for if statements. For statements, $\langle \tilde{\sigma}, s \rangle \tilde{\Downarrow} \tilde{\Sigma}$ says that under the variable complexity mapping $\tilde{\sigma}$, the statement $s$ results in a set of paths with complexity mappings $\tilde{\Sigma}(p)$ for path $p$. We use a period to denote string concatenation (e.g., $1.p$ to prepend $p$ with $1$).

---

[4] `https://github.com/iclr-2022-UcKEodTPtfI/turaco` contains our implementation of TURACO and of the experiments in Sections 2 and 5.

$$\frac{}{\langle\tilde{\sigma},v\rangle\tilde{\Downarrow}(|v|,0)} \qquad \frac{}{\langle\tilde{\sigma},x\rangle\tilde{\Downarrow}\tilde{\sigma}(x)} \qquad \frac{\langle\tilde{\sigma},e_1\rangle\tilde{\Downarrow}(\tilde{v}_1,\tilde{v}_1') \quad \langle\tilde{\sigma},e_2\rangle\tilde{\Downarrow}(\tilde{v}_2,\tilde{v}_2')}{\langle\tilde{\sigma},\mathrm{MUL}(e_1,e_2)\rangle\tilde{\Downarrow}(\tilde{v}_1\cdot\tilde{v}_2,\tilde{v}_1\cdot\tilde{v}_2'+\tilde{v}_2\cdot\tilde{v}_1')}$$

Figure 4: Complexity relation for expressions in TURACO.

$$\frac{\langle\tilde{\sigma},s_1\rangle\tilde{\Downarrow}\tilde{\Sigma}_l \quad \langle\tilde{\sigma},s_2\rangle\tilde{\Downarrow}\tilde{\Sigma}_r}{\langle\tilde{\sigma},\mathrm{if}\,(e>0)\,\{\,s_1\,\}\,\mathrm{else}\,\{\,s_2\,\}\rangle\tilde{\Downarrow}\left\{\mathtt{l}.p\mapsto\tilde{\Sigma}_l(p)\Big|p\in\tilde{\Sigma}_l\right\}\cup\left\{\mathtt{r}.p\mapsto\tilde{\Sigma}_r(p)\Big|p\in\tilde{\Sigma}_r\right\}}$$

Figure 5: Complexity relation for if statements, using a period to denote string concatenation.

To calculate the complexity $\zeta$ as defined in Section 3.3 (for a given input $\beta$ parameter, which represents the scale of the input data) of each path of a program, we use the statement relation to calculate $\tilde{f}'(\beta)$ and $\tilde{f}(0)$. Appendix D.3 presents this rule, along with the theorem that the complexity calculated by this analysis is an upper bound on the complexity as defined in Section 3.3.

## 5 RENDERER DEMONSTRATION

In this section we present a case study of our optimal sampling results and complexity analysis. The program under study is a demonstration 3D renderer (Lettier, 2019), such as forms the core of a graphics rendering pipeline for a movie or 3D game engine (Christensen et al., 2018; Tatarchuk, 2006). Figures 6a and 6b show scenes that the renderer generates. We demonstrate that the sampling and analysis techniques in Sections 3 and 4 consistently result in more accurate surrogates than those trained using baseline distributions (the frequency distribution of paths and the uniform distribution).

Compared to training surrogates on the frequency distribution of paths, optimal path sampling decreases error by $15\%$. Compared to training on the uniform distribution of paths, optimal path sampling decreases error by $47\%$. These improvements in error correspond to perceptual improvements in the generated images, as shown in Figures 6c to 6e.

### 5.1 PROGRAM UNDER STUDY

The full renderer program is a 2750 lines-of-code C++ program, which invokes 38 different GLSL shader programs totaling 2446 lines of code. We learn a surrogate of a section of one core shader, totaling 60 lines of code.[5] Figure 18 in Appendix E presents the code under study.

**Input-output specification.** This program is a shader which assigns colors to pixels in the image based on the scene geometry, materials, lights, and other properties. The program is called for each pixel that is rendered in the image. Each invocation the of program takes as input a set of 11 fixed-size vectors, totaling 35 inputs. The program returns as output a set of 2 fixed-size vectors, totaling 8 outputs.[6] These outputs are two RGBA colors, the first representing the base color of the pixel, and the second representing the color and intensity of a specular map at that pixel.

**Scenes and datasets.** We evaluate the renderer on four different scenes, which we combine into nine different datasets. Figures 6a and 6b present two of the four different scenes under consideration; the four scenes are all combinations of views from the front and top, during the day and night. We combine these scenes into nine datasets: a dataset with each scene, a dataset combining each scene from each angle (front day and front night, top day and top night), a dataset combining each scene from each time of day, and a dataset combining all scenes. Figure 14 in Appendix E presents the full set of scenes under study.

---

[5]Lines 278 through 337 of `https://github.com/lettier/3d-game-shaders-for-begin ners/blob/2970085/demonstration/shaders/fragment/base.frag`.

[6]Our implementation of TURACO extends the interpreter and analysis to vectors, with the resulting complexity being the sum of the complexity of each component of the output vector.

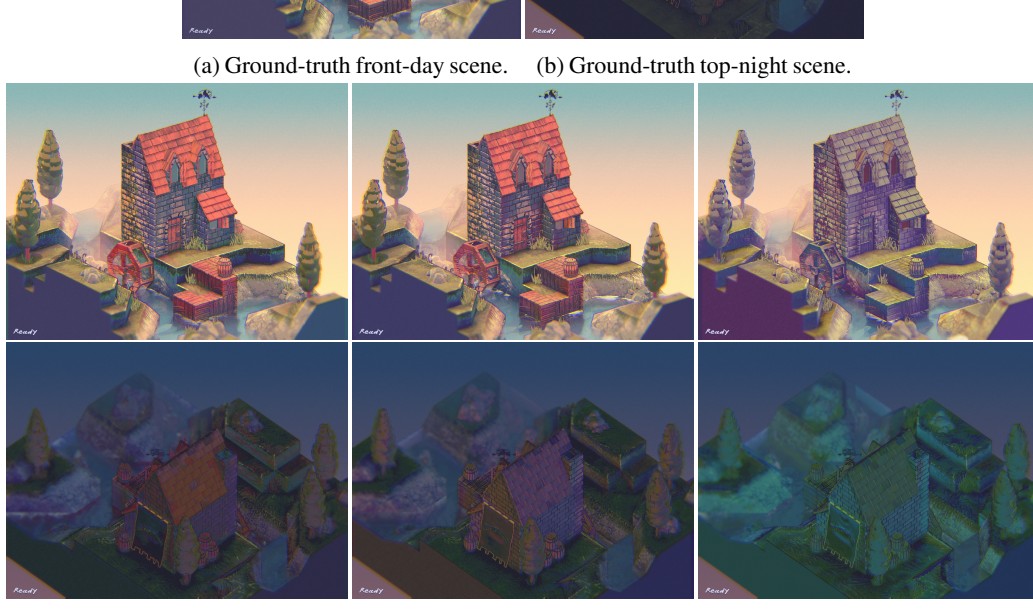

(a) Ground-truth front-day scene.  (b) Ground-truth top-night scene.

(c) Optimal surrogate.  (d) Frequency-based surrogate.  (e) Uniform surrogate.

Figure 6: Ground-truth (top) and surrogate renderings (bottom) of scenes generated by the renderer.

Table 1: Top: the identifier, lines of code, complexity, and description of each path present in our datasets. Bottom: the distribution (abbreviated distr.) of each path across a subset of datasets: the frequency (freq.) of each observed path, and the optimal sampling rate (opt.) of that path.

| Path | | lrrllr | lrrlrl | lrrlrr | lrrrlr | lrrrrl | lrrrrr | rrrllr | rrrlrl | rrrlrr |
|---|---|---|---|---|---|---|---|---|---|---|
| **Lines of Code** | | 17 | 17 | 17 | 18 | 18 | 18 | 17 | 17 | 17 |
| **Complexity** | | 6210 | 5899 | 6369 | 6650 | 6328 | 6814 | 6459 | 6142 | 6621 |
| **Description** | | Twilight Water | Twilight Smoke | Twilight Solids | Nighttime Water | Nighttime Smoke | Nighttime Solids | Daytime Water | Daytime Smoke | Daytime Solids |
| **Dataset** | **Distr.** | **lrrllr** | **lrrlrl** | **lrrlrr** | **lrrrlr** | **lrrrrl** | **lrrrrr** | **rrrllr** | **rrrlrl** | **rrrlrr** |
| Front Day | Freq. | | | | | | | 5.0% | 7.9% | 87.1% |
| | Opt. | | | | | | | 11.0% | 14.7% | 74.4% |
| Top Night | Freq. | 0.16% | 0.06% | 1.2% | 0.3% | 0.1% | 2.4% | 6.3% | 12.9% | 76.5% |
| | Opt. | 0.95% | 0.49% | 3.6% | 1.5% | 0.8% | 5.9% | 10.9% | 17.4% | 58.4% |
| All | Freq. | 0.04% | 0.02% | 0.3% | 1.3% | 2.0% | 13.5% | 4.5% | 8.5% | 69.8% |
| | Opt. | 0.35% | 0.18% | 1.3% | 3.7% | 4.8% | 17.4% | 8.2% | 12.4% | 51.6% |

Table 2: Average change in error across all budgets from using optimal sampling compared to baselines on each dataset (negative means optimal sampling has lower error).

| Baseline | Front Day | Front Night | Top Day | Top Night | Front | Top | Day | Night | All | Mean |
|---|---|---|---|---|---|---|---|---|---|---|
| Frequency | -4% | +6% | +2% | -49% | +3% | -33% | -2% | -13% | -29% | -15% |
| Uniform | -46% | -32% | -36% | -45% | -47% | -64% | -35% | -58% | -53% | -47% |

**Paths.** The program is a conjunction of 48 different paths, 9 of which are exercised by the renderer. The top part of Table 1 presents statistics about the paths under study, showing the identifier (a trace of `l` and `r` characters denoting which branch of each if statement the path takes), the lines of code in the corresponding trace, and the complexity of the corresponding trace according to the analysis in Section 4.2. The paths are broken up into a path for rendering smoke particles from the chimney, water particles in the river, and the solids of the ground and house. Each set of paths is duplicated for twilight, nighttime, and daytime. Within each time of day, the smoke paths are the least complex, followed by water then solids. Across time, twilight paths the least complex, followed by daytime then nighttime.

Table 1 also presents the observed and optimal distributions of paths for each dataset. In general, the twilight paths are rarer than the nighttime paths, which are rarer than the daytime paths: this is because data collection for the nighttime scenes extends through twilight and into the morning. For all datasets, the smoke paths are rarer than the water paths, which are in turn rarer than the solids paths; this is purely due to the number of points observed for each scene.

Appendix E.1 presents code, statistics, and visualizations of all paths in the scene.

## 5.2 SURROGATE TRAINING AND DEPLOYMENT METHODOLOGY

To create and deploy a surrogate of the renderer, we train a surrogate of each path, then create a stratified surrogate which branches on the set of path conditions and applies the corresponding surrogate.

Our goal is to compare the theoretical and empirical errors achieved by training on the optimal sampling distribution against those of baseline distributions. We compare the approaches across different training datasets, different total numbers of training data points, and evaluating across different evaluation sets, all with multiple trials. Full methodological details are presented in Appendix E.2.

## 5.3 SURROGATE ERRORS

Table 2 presents the geometric mean change in error of using optimal sampling compared to each baseline, on each dataset. Across most datasets, optimal path sampling results in lower error than both frequency-based path sampling and uniform path sampling. On datasets with few paths (front-day) and in which all paths are well represented (minimum 5% frequency), the gap is minimal, and frequency-based path sampling matches or outperforms optimal path sampling. On datasets with more and rarer paths (top-night), the gap widens and optimal path sampling outperforms frequency-based path sampling. On all datasets, optimal path sampling outperforms uniform path sampling.

## 5.4 VISUALIZATION

Figure 6 presents the renderings generated by the surrogates for the front-day and the top-night scene. These budgets correspond to the smallest budget that lead to a validation error less than 2%, which was qualitatively chosen as a threshold around which surrogate renders converge on the ground truth.

The top row shows the front day scene using surrogates trained on the front day dataset. In this scene, the optimal and frequency-based surrogates result in similarly accurate renders, with the primary difference being that the frequency-based surrogate is overall more red, which is most notable in the windows and the front face of the house. This similarity is expected given the similar errors observed in Table 2. Uniform sampling results in an inaccurate render, as expected given its high error.

The bottom row show the top night scene using surrogates trained on the dataset combining all scenes. In this scene the optimal surrogate has the most accurate render, as expected given the errors observed in Table 2. The frequency-trained surrogate colors everything (especially the roof and water) slightly more pink. The uniform-trained surrogate colors everything significantly more green.

In sum, the error improvements in Table 2 result in visual improvements in the generated renders.

## 6 CONCLUSION

We present an optimal approach to allocating samples among strata to train stratified neural network surrogates of stratified functions. We also present a programming language, TURACO, in which all programs are learnable stratified functions and a program analysis to determine the complexity of learning surrogates of those programs. Our results take a step towards a cohesive, end-to-end methodology for programming using surrogates of programs.

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

# A    RELATED WORK

In this appendix we survey related work for each contribution.

## A.1    OPTIMAL STRATIFIED SAMPLING

Optimal stratified sampling is a classic area in statistics (Thompson, 2012). Most work in this domain focuses on optimal parameter estimation, and uses stratified sampling to reduce the variance of estimates by ensuring sufficient independent samples are taken from each subpopulation. Our approach is novel in the assumptions we make for training stratified surrogates of programs, and in the specific sample complexity bounds we base our results on.

Santner et al. (2018) survey sampling techniques for computer experiments. Chapter 5.2.3 discusses stratified random sampling in particular, showing optimality criteria for sampling for unbiased estimators. These approaches are generic for minimizing the variance of estimators, and do not consider specifically training a neural network. These approaches also do not consider the different complexity of different strata.

Cortes et al. (2019) present an active learning approach for learning in the regime where the input space is partitioned into separate regions (strata, using our terminology) and a separate hypothesis (surrogate) is trained of each, and derive a similar optimal of allocation of data points. This approach has two primary differences from our approach. First, it is an active learning approach that determines whether or not to query a label of a given data point for an input stream of data points, whereas our approach operates offline. Second, it assumes a different form for sample complexity and derives correspondingly different optimal sampling bounds than ours. The definition of complexity ($\zeta$ in our formalism) that Cortes et al. use is a function of the number of hypotheses in the hypothesis class, the total number of data points used, and the number of data points for a given stratum that have been queried thus far. It is not a function of any complexity metric of the function being learned. Cortes et al.'s approach is thus a better fit when learning stratified functions of unknown complexity (i.e., non-analytic functions) using a finitely sized hypothesis class (not a neural network), and is targeted at the online setting when given a sampler of the overall data distribution but not one for each stratum. Our approach is a better fit when learning stratified analytic functions with neural networks, and is targeted at the offline setting when given a sampler for each stratum.

## A.2    SAMPLE COMPLEXITY PROGRAM ANALYSIS

Program analysis is a broad set of techniques to determine properties of programs (Nielson et al., 1999; Cousot & Cousot, 1977). Our analysis in Section 4.2 is a novel nonstandard interpretation calculating the tilde, combined with a standard implementation of forward-mode automatic differentiation (Wengert, 1964; Griewank & Walther, 2008) and a standard symbolic execution which executes all paths in the program (King, 1976; Cadar et al., 2008).

Bao et al. (2012) present a program analysis that decomposes programs into continuous regions, with the goal of characterizing the sensitivity of each continuous region to input noise. This analysis computes a different notion of complexity than ours, and does not represent the sample complexity of learning a surrogate of each region.

Hoffmann & Hofmann (2010) present a program analysis that calculates the algorithmic complexity of a program. This complexity again does not lead to bounds on the sample complexity of learning a surrogate of the program.

# B    ASSUMPTIONS

Our contributions in Sections 3 and 4 make assumptions about the programs under study, the functions that those programs denote, and the surrogate training algorithms. Here we document these assumptions and note possible failure modes for our techniques.

**Assumptions imported from prior work.**    Our sample complexity results are subject to all assumptions from the prior work that gives the sample complexity bounds for neural networks that we

use (Agarwala et al., 2021). These sample complexity bounds only apply to analytic functions. They further assume that inputs come from the unit sphere; this does not match many practical applications, including those in Section 5. Finally, these sample complexity bounds assume that the neural network under study is a 2-layer, infinitely wide neural network trained with SGD with an infinitely small step size, using a 1-Lipschitz loss function.

Despite these assumptions, Agarwala et al. (2021, Appendix B.2) empirically verify that the sample complexity bounds hold. We also show in Section 2 that the theoretical sample complexity bounds correlate with empirical sample complexity results on three functions.

**Optimal sampling.** The first assumption is that we know the distribution of inputs ahead of time $D(x)$, both in terms of the distribution of strata $D(s_i)$ and the distribution of inputs within a given stratum $D(x|s_i)$. The second assumption is that optimizing the upper bound of the per-stratum loss results in a reasonable optimum for the combined surrogate. The third is the technical assumption we make that $\delta_i = \frac{\delta}{c}$; this is not guaranteed to be optimal.

**Program analysis.** The main assumption here is that Agarwala et al. (2021)'s algebra on tilde functions results in a sufficiently precise upper bound on the tilde. This is not always the case – for instance, this assumption says that `ADD(x, NEG(x))` has a positive tilde (since the tilde of `NEG(x)` is equal to the tilde of `x`), whereas it should have a tilde of 0.

**Stratified surrogates.** We provide sample complexity bounds for constructing stratified surrogates, assuming that for a given program every path is a different function. This assumes both that it is tractable to compute which stratum a given input resides in before applying the surrogate. This also assumes that there are a tractable number of paths, which excludes programs with a large number of if statements or loops.

## C  Proof of Optimal Sampling

This appendix provides the proof for the optimal sampling result in Section 3.

For convenience we duplicate the definitions of learnability. A learnable function is one such that:

$$\forall D,n,\epsilon,\delta.\exists n. \Pr_{\hat{f}\sim tr(f,D,n,\ell)} \left( \mathbb{E}_{x\sim D}\left[\ell\left(\hat{f}(x),f(x)\right)\right] \leq \epsilon \right) \geq 1-\delta \tag{1}$$

Following Arora et al. (2019) and Agarwala et al. (2021) we study functions and learning algorithms for which the relationship between $n$, $\epsilon$, and $\delta$ in Equation (1) is:

$$\exists C.n \leq C\left[\frac{\zeta(f)+\log\left(\delta^{-1}\right)}{\epsilon^2}\right] \tag{2}$$

The optimization problem of our optimal sampling approach is:

$$\operatorname*{argmin}_{\{n_i\}_{i=1}^c \in \mathbb{N}^+} \mathbb{E}_{x\sim D}\left[\ell\left(\hat{f}(x),f(x)\right)\right] \text{ s.t. } \sum_i n_i \leq n \text{ and } \hat{f}_i \sim tr(f_i,D(x|s_i),n_i,\ell) \tag{6}$$

**Theorem 3.1.** *With $\delta_i = \frac{\delta}{c}$, the upper bound of Equation (3) is minimized at:*

$$n_i = n\frac{\left(D(s_i)\sqrt{\zeta(f_i)+\log(c\delta^{-1})}\right)^{\frac{2}{3}}}{\sum_{i=1}^c\left(D(s_i)\sqrt{\zeta(f_i)+\log(c\delta^{-1})}\right)^{\frac{2}{3}}},$$

*Proof.* The task is to find $n_i$ for each surrogate $\hat{f}_i$ that minimize the expectation of error and such that $\sum_i n_i \leq n$. We first separate out the expectation by stratum, and upper bound the error using the sample

complexity assuming for the surrogates:

$$\underset{x\sim D}{\mathbb{E}}\left[\ell\left(\hat{f}(x),f(x)\right)\right]=\underset{s_i\sim\{D(s_i)\}}{\mathbb{E}}\left[\underset{x\sim D(x|s_i)}{\mathbb{E}}\left[\ell\left(\hat{f}(x),f(x)\right)\right]\right]$$

$$\leq\underset{s_i\sim\{D(s_i)\}}{\mathbb{E}}\left[\sqrt{\frac{C}{n}}\left[\zeta(f_i)+\log\left(\delta_i^{-1}\right)\right]\right]$$

$$=\sum_i D(s_i)\sqrt{\frac{C}{n}}\left[\zeta(f_i)+\log\left(\delta_i^{-1}\right)\right]$$

Plugging in $\delta_i$ and writing out the Lagrangian, we have:

$$\mathcal{L}(n_i,\lambda)=\sum_{i=1}^{c}D(s_i)C^{\frac{1}{2}}\sqrt{\zeta(f_i)+\log\frac{c}{\delta}}n_i^{-\frac{1}{2}}+\lambda\left(n-\sum n_i\right)$$

$$\frac{\partial}{\partial n_i}\mathcal{L}=-\frac{1}{2}D(s_i)C^{\frac{1}{2}}\sqrt{\zeta(f_i)+\log\frac{c}{\delta}}n_i^{-\frac{3}{2}}-\lambda=0 \qquad \frac{\partial}{\partial\lambda}\mathcal{L}=n-\sum_i n_i=0$$

$$n_i=\left(\frac{1}{2|\lambda|}D(s_i)C^{\frac{1}{2}}\sqrt{\zeta(f_i)+\log\frac{c}{\delta}}\right)^{\frac{2}{3}} \qquad n=\sum_i n_i$$

$$n=\sum_i\left(\frac{1}{2|\lambda|}D(s_i)C^{\frac{1}{2}}\sqrt{\zeta(f_i)+\log\frac{c}{\delta}}\right)^{\frac{2}{3}}$$

$$|\lambda|=\left(\frac{1}{n}\sum_i\left(\frac{1}{2}D(s_i)C^{\frac{1}{2}}\sqrt{\zeta(f_i)+\log\frac{c}{\delta}}\right)^{\frac{2}{3}}\right)^{\frac{3}{2}}$$

$$n_i=\left(\frac{1}{n}\sum_i\left(\frac{1}{2}D(s_i)C^{\frac{1}{2}}\sqrt{\zeta(f_i)+\log\frac{c}{\delta}}\right)^{\frac{2}{3}}\right)^{-1}\left(\frac{1}{2}D(s_i)C^{\frac{1}{2}}\sqrt{\zeta(f_i)+\log\frac{c}{\delta}}\right)^{\frac{2}{3}}$$

$$n_i=n\frac{\left(D(s_i)\sqrt{\zeta(f_i)+\log\frac{c}{\delta}}\right)^{\frac{2}{3}}}{\sum_i\left(D(s_i)\sqrt{\zeta(f_i)+\log\frac{c}{\delta}}\right)^{\frac{2}{3}}}$$

Because the objective is convex in $n_i$, this is the optimal solution. $\qquad\square$

## D    FULL TURACO SEMANTICS

This appendix presents the full semantics and analysis for TURACO. Figure 7 re-presents TURACO.

### D.1    NORMAL EVALUATION SEMANTICS

Figure 8 presents the big-step evaluation relation for expressions in TURACO. The expression relation $\langle\sigma,e\rangle\Downarrow v$ says that under variable store $\sigma$ (assigning values to all variables in $e$), the expression $e$ evaluates to value $v$.

$$p::=\text{fun}\,(x+)\,\{\,s\,\}\,\text{return}\,x$$
$$s::=s;s\,|\,x:=e\,|\,\text{if}\,(e>0)\,\{\,s\,\}\,\text{else}\,\{\,s\,\}\,|\,\text{skip}$$
$$e::=x\,|\,v\,|\,b(e,e)\,|\,u(e)$$
$$b::=\text{ADD}\,|\,\text{MUL}$$
$$u::=\text{NEG}\,|\,\text{SIN}\,|\,\text{EXP}\,|\,\text{LOG1P}$$
$$x::=\text{set of variable names}$$
$$v::=\text{set of floating point values}$$

Figure 7: Syntax of TURACO.

$$\overline{\langle \sigma,v \rangle \Downarrow |v|} \qquad \overline{\langle \sigma,x \rangle \Downarrow \sigma(x)} \qquad \frac{\langle \sigma,e \rangle \Downarrow v}{\langle \sigma,\mathrm{NEG}(e) \rangle \Downarrow -v} \qquad \frac{\langle \sigma,e \rangle \Downarrow v}{\langle \sigma,\mathrm{SIN}(e) \rangle \Downarrow \sin(a)}$$

$$\frac{\langle \sigma,e \rangle \Downarrow v}{\langle \sigma,\mathrm{EXP}(e) \rangle \Downarrow \exp(v)} \qquad \frac{\langle \sigma,e \rangle \Downarrow v \qquad -1 < v \le 1}{\langle \sigma,\mathrm{LOG1P}(e) \rangle \Downarrow \log(1+v)} \qquad \frac{\langle \sigma,e_1 \rangle \Downarrow v_1 \qquad \langle \sigma,e_2 \rangle \Downarrow v_2}{\langle \sigma,\mathrm{ADD}(e_1,e_2) \rangle \Downarrow v_1 + v_2}$$

$$\frac{\langle \sigma,e_1 \rangle \Downarrow v_1 \qquad \langle \sigma,e_2 \rangle \Downarrow v_2}{\langle \sigma,\mathrm{MUL}(e_1,e_2) \rangle \Downarrow v_1 \cdot v_2}$$

Figure 8: Big-step evaluation relation for expressions in TURACO.

$$\frac{\langle \sigma,s_1 \rangle \Downarrow \sigma' \qquad \langle \sigma',s_2 \rangle \Downarrow \sigma''}{\langle \sigma,s_1;s_2 \rangle \Downarrow \sigma''} \qquad \frac{\sigma \vdash e \rangle \Downarrow v}{\langle \sigma,x := e \rangle \Downarrow \sigma[x \mapsto v]} \qquad \frac{\langle \sigma,e \rangle \Downarrow v \qquad v > 0 \qquad \langle \sigma,s_1 \rangle \Downarrow \sigma_l}{\langle \sigma,\mathrm{if}\,(e > 0)\,\{\,s_1\,\}\,\mathrm{else}\,\{\,s_2\,\} \rangle \Downarrow \sigma_l}$$

$$\frac{\langle \sigma,e \rangle \Downarrow v \qquad v \le 0 \qquad \langle \sigma,s_2 \rangle \Downarrow \sigma_r}{\langle \sigma,\mathrm{if}\,(e > 0)\,\{\,s_1\,\}\,\mathrm{else}\,\{\,s_2\,\} \rangle \Downarrow \sigma_r}$$

Figure 9: Big-step evaluation relation for statements.

$$\frac{\langle \sigma,s \rangle \Downarrow \sigma}{\langle \sigma,\mathrm{fun}\,(x_0,x_1...,x_n)\,\{\,s\,\}\,\mathrm{return}\,x \rangle \Downarrow \sigma(x)}$$

Figure 10: Big-step evaluation relation for programs.

Figure 9 presents the big-step evaluation relation for statements in TURACO. The statement relation $\langle \sigma,s \rangle \Downarrow \sigma'$ says that under variable store $\sigma$, the statement $s$ evaluates to a new variable store $\sigma'$.

Figure 10 presents the big-step evaluation relation for TURACO programs. The program relation $\langle \sigma,\mathrm{fun}\,(x_0,x_1...,x_n)\,\{\,s\,\}\,\mathrm{return}\,x \rangle \Downarrow v$ says that under variable store $\sigma$ (assigning values to all $x_i$) the program evaluates to value $v$.

## D.2 COMPLEXITY ANALYSIS

Figure 11 presents the full complexity relation used to calculate the complexity for all expressions. The relation $\langle \tilde{\sigma},[e] \rangle \tilde{\Downarrow} (\tilde{v},\tilde{v}')$ says that under the variable complexity mapping $\tilde{\sigma}$ (mapping variables to tuples with their respective tilde and tilde derivative), the expression $e$ has $\tilde{e} \le \tilde{v}$ and $\tilde{e}' \le \tilde{v}'$.

Figure 12 presents the complexity relation for statements. For statements, the relation $\langle \tilde{\sigma},s \rangle \tilde{\Downarrow} \tilde{\Sigma}$ says that under the variable complexity mapping $\tilde{\sigma}$, the statement $s$ results in a set of paths with variable complexity mappings $\tilde{\Sigma}(p)$ for path $p$.

Figure 13 presents the complexity $\zeta$ (for a given input $\beta$ parameter) of each path of a program.

## D.3 ANALYSIS CORRECTNESS

This appendix presents the statement of correctness for the TURACO complexity analysis.

**Theorem 1.** *Given:*
$$\zeta_\beta[\mathrm{fun}\,(x_0,x_1,...,x_n)\,\{\,s\,\}\,\mathrm{return}\,x] \le \{p \mapsto \tilde{v}_p\}$$
*then for all paths $p$, $\zeta_\beta(f) \le \tilde{v}_p$, where $f$ is the function denoted by the standard execution of the path $p$ through $s$.*

*Proof.* The proof is by induction using Fact 1 (p. 17) and Lemma 7 (p. 22) in Agarwala et al. (2021). □

$$\overline{\langle\tilde\sigma,v\rangle\tilde\Downarrow(|v|,0)} \qquad \overline{\langle\tilde\sigma,x\rangle\tilde\Downarrow\tilde\sigma(x)} \qquad \frac{\langle\tilde\sigma,e\rangle\tilde\Downarrow(\tilde v,\tilde v')}{\langle\tilde\sigma,\mathrm{NEG}(e)\rangle\tilde\Downarrow(\tilde v,\tilde v')} \qquad \frac{\langle\tilde\sigma,e\rangle\tilde\Downarrow(\tilde v,\tilde v')}{\langle\tilde\sigma,\mathrm{SIN}(e)\rangle\tilde\Downarrow(\sinh(\tilde v),\tilde v'\cosh(\tilde v))}$$

$$\frac{\langle\tilde\sigma,e\rangle\tilde\Downarrow(\tilde v,\tilde v')}{\langle\tilde\sigma,\mathrm{EXP}(e)\rangle\tilde\Downarrow(\exp(\tilde v),\tilde v'\exp(\tilde v'))} \qquad \frac{\langle\tilde\sigma,e\rangle\tilde\Downarrow(\tilde v,\tilde v') \qquad 0\tilde\Downarrow\tilde v<2}{\langle\tilde\sigma,\mathrm{LOG1P}(e)\rangle\tilde\Downarrow\left(\log(2-\tilde v),\dfrac{\tilde v'}{2-\tilde v}\right)}$$

$$\frac{\langle\tilde\sigma,e_1\rangle\tilde\Downarrow(\tilde v,\tilde v') \qquad \langle\tilde\sigma,e_2\rangle\tilde\Downarrow(\tilde v_2,\tilde v_2')}{\langle\tilde\sigma,\mathrm{ADD}(e_1,e_2)\rangle\tilde\Downarrow(\tilde v+\tilde v_2,\tilde v'+\tilde v_2')} \qquad \frac{\langle\tilde\sigma,e_1\rangle\tilde\Downarrow(\tilde v,\tilde v') \qquad \langle\tilde\sigma,e_2\rangle\tilde\Downarrow(\tilde v_2,\tilde v_2')}{\langle\tilde\sigma,\mathrm{MUL}(e_1,e_2)\rangle\tilde\Downarrow(\tilde v\cdot\tilde v_2,\tilde v\cdot\tilde v_2'+\tilde v_2\cdot\tilde v')}$$

Figure 11: Complexity relation for expressions in TURACO.

$$\frac{\langle\tilde\sigma,s_1\rangle\tilde\Downarrow\tilde\Sigma' \qquad \langle\tilde\sigma',s_2\rangle\tilde\Downarrow\tilde\Sigma''}{\langle\tilde\sigma,s_1;s_2\rangle\tilde\Downarrow\tilde\Sigma''} \qquad \frac{\tilde\sigma\vdash\zeta[e]\le(\tilde v,\tilde v')}{\langle\tilde\sigma,x:=e\rangle\tilde\Downarrow\{\cdot\mapsto\tilde\sigma[x\mapsto(\tilde v,\tilde v')]\}}$$

$$\frac{\langle\tilde\sigma,s_1\rangle\tilde\Downarrow\tilde\Sigma_l \qquad \langle\tilde\sigma,s_2\rangle\tilde\Downarrow\tilde\Sigma_r}{\langle\tilde\sigma,\mathrm{if}\,(e>0)\,\{\,s_1\,\}\,\mathrm{else}\,\{\,s_2\,\}\rangle\tilde\Downarrow\left\{\mathtt{l}.p\mapsto\tilde\Sigma_l(p)\Big|p\in\tilde\Sigma_l\right\}\cup\left\{\mathtt{r}.p\mapsto\tilde\Sigma_r(p)\Big|p\in\tilde\Sigma_r\right\}}$$

Figure 12: Complexity relation for statements.

$$\frac{\begin{array}{cc}\langle[x_i\mapsto(0,1)],s\rangle\tilde\Downarrow\tilde\Sigma_0 & \langle[x_i\mapsto(\beta,1)],s\rangle\tilde\Downarrow\tilde\Sigma_\beta \\ \forall p.\tilde\Sigma_0(p)(x)=(f_{p,0},f_{p,0}') & \forall p.\tilde\Sigma_\beta(p)(x)=(\tilde v_{p,\beta},\tilde v_{p,\beta}')\end{array}}{\zeta_\beta[\mathrm{fun}\,(\{\{x_i\})\,\{\,s\,\}\,\mathrm{return}\,x]\le\left\{p\mapsto\left(\tilde v_{p,0}+\tilde v_{p,\beta}'\right)^2\Big|p\in\tilde\Sigma_0\right\}}$$

Figure 13: Complexity relation for full programs.

## E  EXTENDED RENDERER DEMONSTRATION

This appendix provides further details on the evaluation on the renderer program in Section 5.

Figure 18 presents the full code for the renderer program under study.[7] Figure 14 presents the full set of scenes used in our evaluation.

This program is a good candidate to train a surrogate of for several reasons. First, it is an approximable program: as long as the outputs of a surrogate of the program are sufficiently close to the ground-truth outputs, the generated image will be perceptually indistinguishable. Second, its paths are all determined by *uniform* input variables, variables in GLSL that are constant across each invocation of the shader. This means that relative to the cost of executing the program, it is cheap to determine which path a given input induces in the program (and thus which surrogate to apply). Third, its execution environment is well suited to be replaced with a neural network, since the original program itself performs batch processing on a GPU.

### E.1  PATHS

Table 3 presents statistics about the entire set of paths paths under study.

Figure 16 presents a side-by-side comparison of traces of two paths: `lrrlrl` (twilight smoke), the least complex path in the dataset, and `lrrrrr` (nighttime solids), the most complex path in the dataset.

---

[7]The original program is written in GLSL. We present a semantically equivalent translation (preserving all paths) of the program to TURACO for simplicity of presentation.

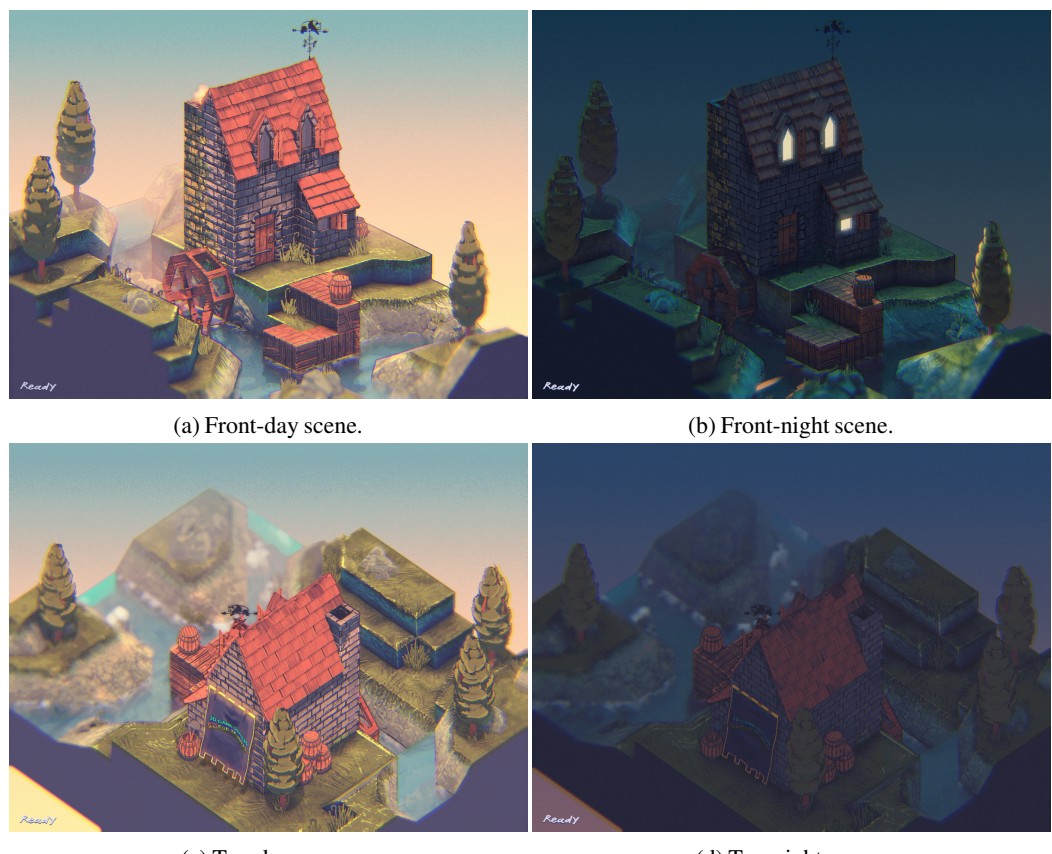

(a) Front-day scene.        (b) Front-night scene.

(c) Top-day scene.        (d) Top-night scene.

Figure 14: Ground-truth scenes generated by the renderer.

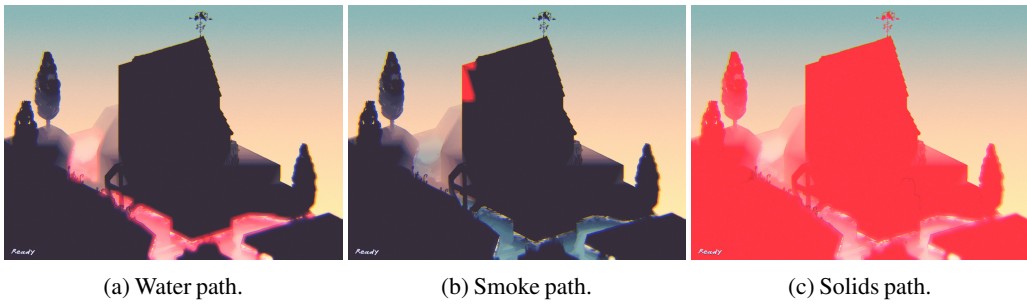

(a) Water path.    (b) Smoke path.    (c) Solids path.

Figure 15: Daytime scene with each different path highlighted red, and all others black.

Two factors lead to the twilight smoke path being less complex than the nighttime solids path. First, the twilight smoke path computes one lighting factor from a constant (Figure 16a, line 14) rather than from a function of the sun's position (Figure 16b, lines 13-14). Second, the twilight smoke path does not have any specular output (Figure 16a, line 18), unlike the nighttime solid output (Figure 16b, line 18).

Figure 15 shows side-by-side comparisons of the three classes of paths: water, smoke, and solids. In each of these images, one path returns red for all pixels, while the other paths return black for all pixels. The base scene is the front daytime scene in Figure 19a.

Table 3: Top: the identifier, lines of code, complexity, and description of each path present in our datasets. Bottom: the distribution (abbreviated distr.) of each path across each dataset: the frequency (freq.) of each observed path, and the optimal sampling rate (opt.) of that path.

| Path | | lrrllr | lrrlrl | lrrlrr | lrrrlr | lrrrrl | lrrrrr | rrrllr | rrrlrl | rrrlrr |
|---|---|---|---|---|---|---|---|---|---|---|
| **Lines of Code** | | 17 | 17 | 17 | 18 | 18 | 18 | 17 | 17 | 17 |
| **Complexity** | | 6210 | 5899 | 6369 | 6650 | 6328 | 6814 | 6459 | 6142 | 6621 |
| **Description** | | Twilight Water | Twilight Smoke | Twilight Solids | Nighttime Water | Nighttime Smoke | Nighttime Solids | Daytime Water | Daytime Smoke | Daytime Solids |
| **Dataset** | **Distr.** | **lrrllr** | **lrrlrl** | **lrrlrr** | **lrrrlr** | **lrrrrl** | **lrrrrr** | **rrrllr** | **rrrlrl** | **rrrlrr** |
| Front Day | Freq. | | | | | | | 5.0% | 7.9% | 87.1% |
| | Opt. | | | | | | | 11.0% | 14.7% | 74.4% |
| Front Night | Freq. | | | | 5.0% | 7.9% | 51.4% | | | 35.6% |
| | Opt. | | | | 9.3% | 12.4% | 44.1% | | | 34.2% |
| Top Day | Freq. | | | | | | | 6.7% | 13.1% | 80.1% |
| | Opt. | | | | | | | 12.8% | 19.7% | 67.4% |
| Top Night | Freq. | 0.16% | 0.06% | 1.2% | 0.3% | 0.1% | 2.4% | 6.3% | 12.9% | 76.5% |
| | Opt. | 0.95% | 0.49% | 3.6% | 1.5% | 0.8% | 5.9% | 10.9% | 17.4% | 58.4% |
| Front | Freq. | | | | 2.5% | 4.0% | 25.7% | 2.5% | 4.0% | 61.4% |
| | Opt. | | | | 5.6% | 7.5% | 26.7% | 5.5% | 7.4% | 47.2% |
| Top | Freq. | 0.08% | 0.03% | 0.6% | 0.2% | 0.1% | 1.2% | 6.5% | 13.0% | 78.3% |
| | Opt. | 0.62% | 0.32% | 2.4% | 1.0% | 0.5% | 3.8% | 11.6% | 18.2% | 61.5% |
| Day | Freq. | | | | | | | 5.9% | 10.5% | 83.6% |
| | Opt. | | | | | | | 11.9% | 17.3% | 70.7% |
| Night | Freq. | 0.08% | 0.03% | 0.6% | 2.7% | 4.0% | 26.9% | 3.1% | 6.5% | 56.1% |
| | Opt. | 0.53% | 0.28% | 2.0% | 5.6% | 7.2% | 26.2% | 6.1% | 9.8% | 42.3% |
| All | Freq. | 0.04% | 0.02% | 0.3% | 1.3% | 2.0% | 13.5% | 4.5% | 8.5% | 69.8% |
| | Opt. | 0.35% | 0.18% | 1.3% | 3.7% | 4.8% | 17.4% | 8.2% | 12.4% | 51.6% |

```
1  fun (...) {
2    sunPosition := sin(mul(sunPosition[0],
        ↪ mul(pi, 0.0055556)))
...
14
15   emission := mul(emission, 0.1)
16   out0rgb := add(..., emission)
17   out0a := diffuseColor[3]
18   out1a := diffuseColor[3]
19   out1rgb := [0.0, 0.0, 0.0]
20 } return (out0rgb, out0a, out1rgb, out1a)
```

```
1  fun (...) {
2    sunPosition := sin(mul(sunPosition[0],
        ↪ mul(pi, 0.0055556)))
...
14   sunPositionPow := exp(mul(log(add(sunPosition, -1.0)), 0.4))
15   emission := mul(emission, sunPositionPow)
16   out0rgb := add(..., emission)
17   out0a := diffuseColor[3]
18   out1a := diffuseColor[3]
19   out1rgb := [specular[0],specular[1],specular[2]]
20 } return (out0rgb, out0a, out1rgb, out1a)
```

(a) Twilight smoke (lrrlrl) trace.      (b) Nighttime solids (lrrrrr) trace.

Figure 16: Side-by-side comparison of the trace of the least complex path (lrrlrl: twilight smoke) and that of the most complex path (lrrrrr: nighttime solids), with differences highlighted.

## E.2 ARCHITECTURE AND HYPERPARAMETER DETAILS

For the design of each surrogate, we use a simple MLP architecture with a single hidden layer of 512 units and a ReLU activations. This architecture matches that of Agarwala et al. (2021), except using 512 rather than 1000 hidden units (we found that the accuracy of each path surrogate plateaued by 512 hidden units). We train the surrogate using the Adam optimizer with a learning rate of 0.0001 and a batch size of 256 for 50,000 steps. We run 5 trials of all experiments, and report the mean error.

## E.3 FULL RESULTS

Figure 17 presents the error of surrogates on each dataset. Each plot shows the error for a different dataset. Each plot has three different lines, respectively showing the error of each surrogate training distribution (optimal, frequency-based, and uniform). Each x axis is the total training data budget. Each y axis is the error of the resulting stratified surrogate.

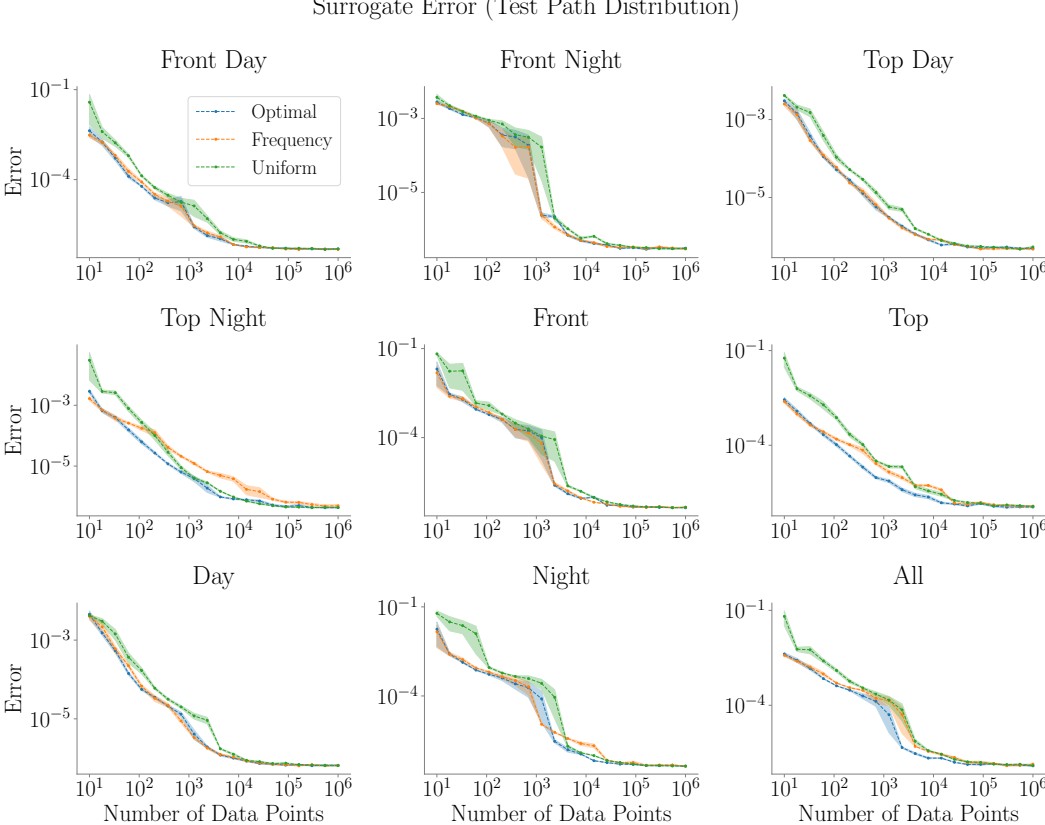

Figure 17: Errors of stratified surrogates of each dataset.

Figures 19 and 20 present larger versions of the renders in Section 5.4 for ease of comparison. The training budgets for these surrogates are 61 samples for the front-day surrogate and 206 samples for the top-night surrogate.

```
1   fun (rimLight[4], isCelShadingEnabled[2], sunPosition[2], gamma[2],
        ↪ worldNormal[3], ssao[3], diffuseColor[4], diffuse[4], specular[4],
        ↪ emission[3], isWater[2], isParticle[2]) {
2    sunPosition = sin(mul(sunPosition[0], mul(pi, 0.0055556)))
3    sunMixFactor = add(0.5, neg(mul(sunPosition, 0.5)))
4    ambientCoolBase = exp(mul(log(add((0.302, 0.451, 0.471), -1)),
        ↪ gamma[0]))
5    ambientWarmBase = exp(mul(log(add((0.765, 0.573, 0.400), -1)),
        ↪ gamma[0]))
6
7    if (0.5 > sunMixFactor) {
8     ambientCool = mul(ambientCoolBase, 0.5)
9     ambientWarm = mul(ambientWarmBase, 0.5)
10   } else {
11    ambientCool = mul(ambientCoolBase, sunMixFactor)
12    ambientWarm = mul(ambientWarmBase, sunMixFactor)
13   }
14
15   if (0 > sunMixFactor) {
16    skyLight = ambientCool
17    groundLight = ambientWarm
18   } else {
19    if (sunMixFactor > 1) {
20     skyLight = ambientWarm
21     groundLight = ambientCool
22    } else {
23     skyLight = add(mul(ambientCool, add(1, neg(sunMixFactor))),
        ↪ mul(ambientWarm, sunMixFactor))
24     groundLight = add(mul(ambientWarm, add(1, neg(sunMixFactor))),
        ↪ mul(ambientCool, sunMixFactor))
25    }
26   }
27
28   worldNormalMixFactor = mul(0.5, add(1.0, worldNormal[2]))
29   ambientLight = add(mul(groundLight, add(1, neg(worldNormalMixFactor))),
        ↪ mul(skyLight, worldNormalMixFactor))
30   ambient = mul(ambientLight, mul((diffuseColor[0], diffuseColor[1],
        ↪ diffuseColor[2]), ssao))
31   if (0.00316228 > sunPosition) {
32    emission = mul(emission, 0.1)
33   } else {
34    sunPositionPow = exp(mul(log(add(sunPosition, -1)), 0.4))
35    emission = mul(emission, sunPositionPow)
36   }
37
38   out0rgb = add(add((ambient[0], ambient[1], ambient[2]), (diffuse[0],
        ↪ diffuse[1], diffuse[2])), add((rimLight[0], rimLight[1],
        ↪ rimLight[2]), emission))
39
40   if (isWater[0] > 0) {
41     out0a = 0
42   } else {
43     out0a = diffuseColor[3]
44   }
45   out1a  = diffuseColor[3]
46
47   if (isParticle[0] > 0) {
48    out1rgb = (0,0,0)
49   } else {
50    out1rgb = (specular[0], specular[1], specular[2])
51   }
52  } return (out0rgb, out0a, out1rgb, out1a)
```

Figure 18: Full code for the renderer case study.

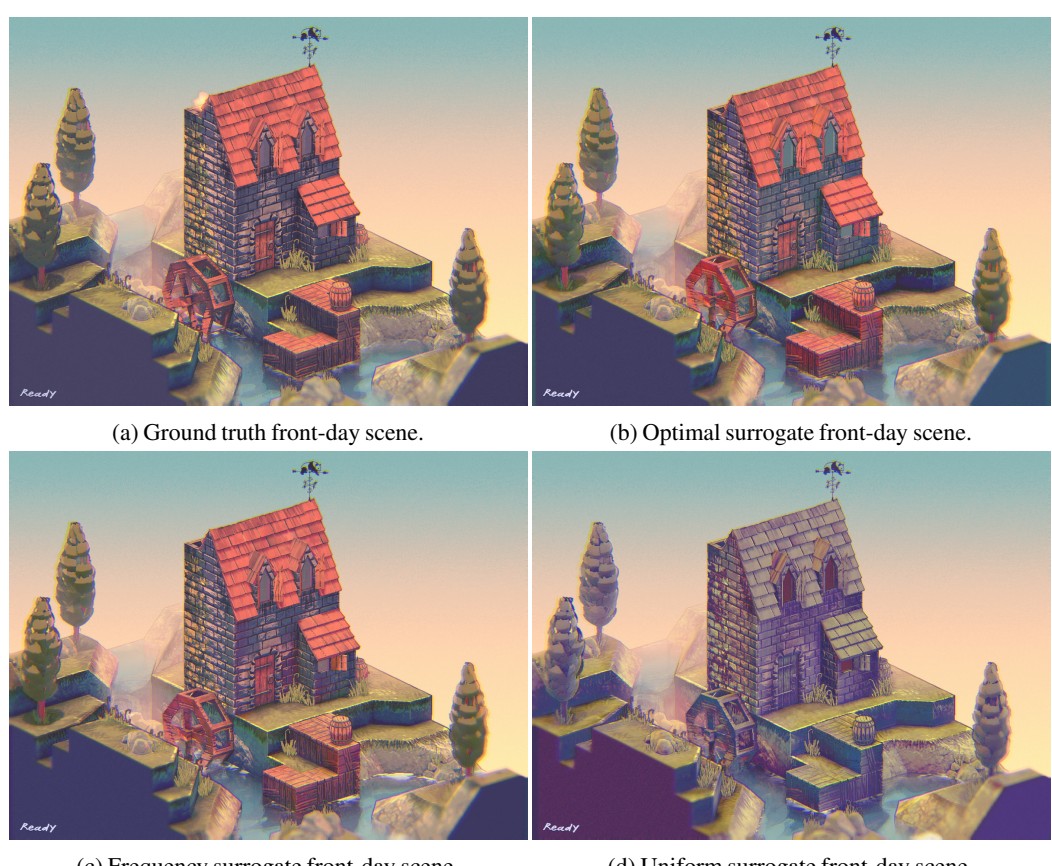

(a) Ground truth front-day scene.

(b) Optimal surrogate front-day scene.

(c) Frequency surrogate front-day scene.

(d) Uniform surrogate front-day scene.

Figure 19: Front-day scene (ground truth in the top left; surrogates in others).

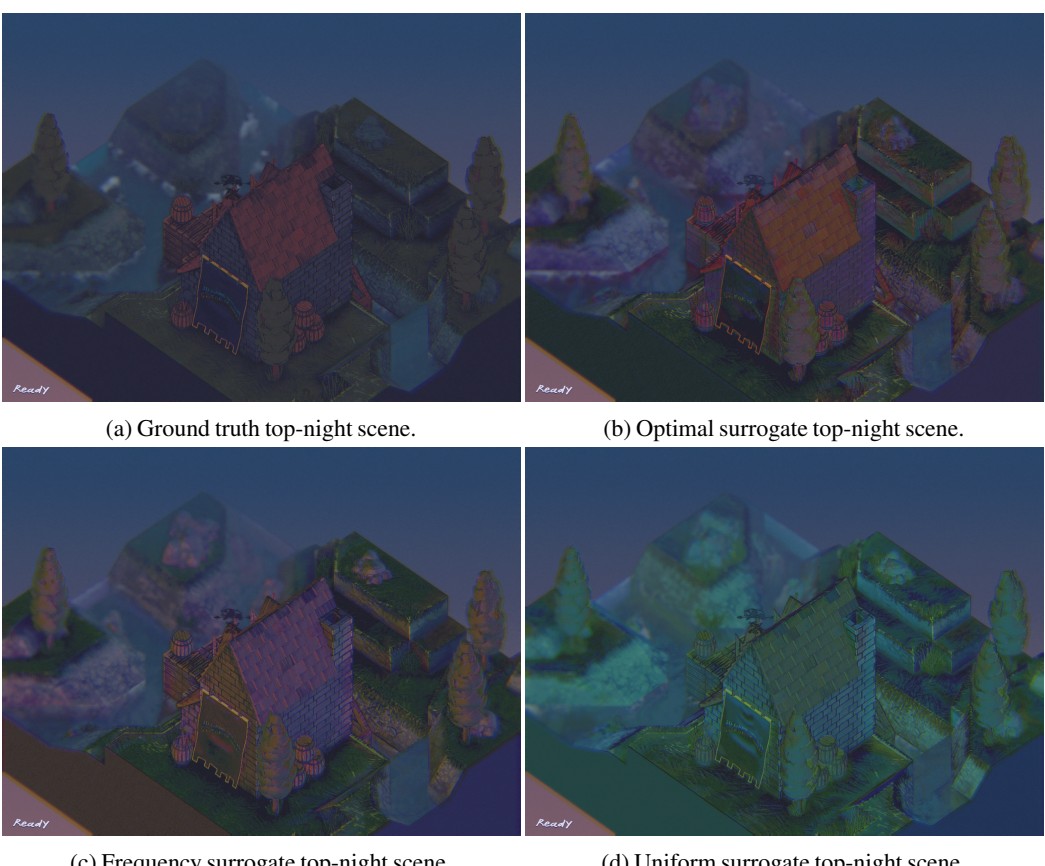

(a) Ground truth top-night scene.      (b) Optimal surrogate top-night scene.

(c) Frequency surrogate top-night scene.    (d) Uniform surrogate top-night scene.

Figure 20: Top-night scene (ground truth in the top left; surrogates from the "all" dataset in others).

