# OpenReview forum: "Optimal Data Sampling for Training Neural Surrogates of Programs"
_ICLR.cc/2023/Conference — Submitted to ICLR 2023_

### Official Review · Reviewer_RLWc · 2022-10-22

**Confidence:** 4
**Correctness:** 2
**Technical Novelty And Significance:** 3
**Empirical Novelty And Significance:** 3
**Recommendation:** 3

**Clarity, Quality, Novelty And Reproducibility:**


### Clarity

The paper is well-written and fairly easy to follow despite the many different
elements of the topic which are combined. Section 2 with the example is
especially helpful.

### Quality

I found the paper to be of high quality.

### Novelty

The paper has novel and interesting results, and it addresses novelty directly
in Appendix B.

### Reproducibility

This is a concern for the paper. I believe that in order for the results to be
reproducible, the implementations of the Turaco programming language and the
methodology to compute the complexity bounds of the traces should be made
public.


**Strength And Weaknesses:**


### Strengths

The paper presents a solid theoretical analysis with proofs, which is also
verified empirically on a real-world program.

### Weaknesses and Questions

The paper works with upper bounds both for Theorem 3.1 and when determining the
complexity of a trace of the program, so we substitute upper bounds of the
complexities of traces into an estimate arrived by minimizing the upper bound of
the error. Even though the authors present empirically verified gains compared
to the baselines, it would be good to know something about how tight these
bounds are.

The complexities of the traces in Table 1 are very similar to each other in
magnitude, probably because the branches in the program don't differ that much
in complexity. It would be good to see a program where the complexities are more
skewed. The path frequencies are the opposite of this: they are very skewed
towards rrrlrr (the rightmost one). Here it would be good to see a less skewed
distribution.

The Turaco programming language lacks iterative constructs like loops or
recursion, which is very important for programming. Could they also be included,
and how difficult would it be?

I think that the original C++ code of the shader (for which there is a GitHub
link on page 7) should be included in the Appendix as it may change on GitHub.

The captions of Table 1 and 2 should be below the tables.

**Summary Of The Paper:**

The paper presents a methodology to optimally sample datasets to train neural
network based surrogates of programs written in Turaco, a programming language
introduced by the authors. The programs are represented as stratified functions,
which are functions that behave differently in different regions (strata) of the
input space (these correspond to different traces of the program according to
the branches chosen during execution). An independent neural network surrogate
is trained for each strata. The question the paper tackles is how to sample the
dataset to train these networks, that is, given a budget of how many samples we
have, how many samples to allocate to the training of each strata.

The authors determine the optimal number of samples $n_i$ for the strata by
minimizing the upper bound of the error of the learned stratified surrogate of
the stratified function, with the constraint that the sum of the number of
samples must be equal to the budget (Theorem 3.1).

To compute the number of samples using this theorem, the complexity of the
component functions $\zeta(f_i)$ is required. To this end, the authors introduce
a programming language called Turaco, in which all programs denote learnable
stratified functions. They also provide the semantics and a complexity analysis
for the language which makes it possible to calculate an upper bound on the
complexity of traces $\zeta$, and so the number of samples $n_i$.

The theoretical results are verified empirically on a 60 line shader program
which is part of a larger 3D renderer. The authors find considerable reductions
in error compared to the baselines of uniform and frequency-based sampling.


**Summary Of The Review:**

Overall I found the paper to be a significant contribution, well-written, and
convincing. Reproducibility is a concern, but that could be easily addressed.

I did not verify the proofs in the Appendix.

------------------------------------------------------------------------------

Update:

Similarly to reviewer #2, I read the comments of reviewer #1 and their
discussion with the authors. I agree with reviewer #1's points that calling the
method "optimal sampling" and the method converging to uniform sampling are
significant issues, so I'm decreasing my score.

---

> ### Author Response · Authors · 2022-11-18
> **Response to Reviewer RLWc**
>
> > it would be good to know something about how tight these bounds are.
>
> Please see [the general response](https://openreview.net/forum?id=UcKEodTPtfI&noteId=ycRp1OZSni) for a discussion of this particular bound.
>
> > It would be good to see a program where the complexities are more skewed. The path frequencies are the opposite of this… [h]ere it would be good to see a less skewed distribution.
>
> Section 2 presents an example with more skewed complexities (specifically, the complexities of Daytime, Twilight, and Nighttime are respectively 29, 2.42, and 0.02). Section 2 also presents an example with somewhat less skewed frequency distribution, as does the Front Night scene (Table 3 in Appendix E).
>
> > The Turaco programming language lacks iterative constructs like loops or recursion, which is very important for programming. Could they also be included, and how difficult would it be?
>
> Please see [the general response](https://openreview.net/forum?id=UcKEodTPtfI&noteId=ycRp1OZSni) for a discussion of Turaco as a programming language.
>
> > I think that the original C++ code of the shader (for which there is a GitHub link on page 7) should be included in the Appendix as it may change on GitHub.
>
> > the implementations of the Turaco programming language and the methodology to compute the complexity bounds of the traces should be made public.
>
> Please see [the general response](https://openreview.net/forum?id=UcKEodTPtfI&noteId=ycRp1OZSni) above for a link to the source code for the implementation and experiments.

---

### Official Review · Reviewer_E5p2 · 2022-10-24

**Confidence:** 4
**Correctness:** 3
**Technical Novelty And Significance:** 2
**Empirical Novelty And Significance:** 2
**Recommendation:** 3

**Clarity, Quality, Novelty And Reproducibility:**

Clarity: The paper is mostly clear, well-written, and easy to follow.

Quality: The quality of this paper looks reasonable and they achieve more accurate surrogates than existing works.

Novelty: The novelty of the proposed approach is medium. But they present a new programming language, which is quite novel.

Reproducibility: The author has uploaded their code in the github, so this work is reproducibility.

**Strength And Weaknesses:**

Strength:
The authors proposed an optimal allocation approach, which allocates different number of samples by considering different complexity of the path in the program.
In addition, they also presented a new programming language to realize the proposed method.


Major comments:
Page 2 "Opitmal sampling". The authors mentioned that "Using neural network sample complexity bounds for learning analytic functions". The reviewer is wondering are there other complex bounds that can be used to learn analytic functions? If so, what are the differences between these bounds, and why did the authors choose this bound?

Page 3 "Optimal path sampling". The authors mentioned that "Using this bound (as implemented by our TURACO analysis described in Section 4.2), we determine that the twilight path takes 1.5× as many samples to train a surrogate to a given error as the nighttime path, and the daytime path requires5× as many samples." The reviewer is confused about this result. The frequency of these three paths is in the order: nighttime>daytime>twilight, but the complexity order is: nighttime <  twilight <  daytime. Could you explain how this happened?

Page 4, Figure 2 (b). It seems that when the total data size is small (around 10^1) the error decreases faster under the frequency sampling. Could you provide a possible reason for this?

Page 4, "Training methodology".  Is the proposed method sensitive to these parameters?  How do these parameters influence the result?

Page 4, Section 3 "A given function f is probably approximately correctly learnable...". Please provide the definition of "probably approximately correctly learnable".

Page 6. "For a 2-layer neural network trained with stochastic gradient descent..." Instead of stochastic gradient decent, can other methods be used?

Page 6, section 4. The motivation to create this new language "TuARO" is not clear. Specifically, why not use other existing languages, e.g. C++, python? Compared with these existing languages, what are the advantages of the presented language?

Page 12, A1. Could you please list the pros and cons of your offline method compared with the online method in Coetes et al (2019)? In addition, could you explain how the different choice on the sample complexity affects the result?

Page 13 "Optimal sampling". It is suggested that the restrictiveness of the assumptions should be discussed.

**Summary Of The Paper:**

This paper is concerned with the problem of developing surrogates of programs. A major difficulty in this context is how train surrogates for programs with control flow. To overcome this challenge, this paper represents the program as a stratified function and uses stratified surrogates to model such functions. To ensure the training accuracy, an optimal allocation approach allocating different number of training samples by considering the complexity of the different paths of the program is proposed. A new programming language is presented to realize the proposed method. Finally, a demonstration 3D renderer is used for case study to verify the effectiveness of the proposed method.

**Summary Of The Review:**

This paper is well written in general, and the simulation is nice.
The contribution is interesting and enough, but the method seems not very novel.
The motivation for the presented programming language should be improved.

---------------------------------
After reading reviewer #1's comments and discussions, I tend to decrease the score. Although interesting, the paper seems not very rigorous in defining "optimal sampling", "formulation", and the contributions of the paper seem not significant.

---

> ### Author Response · Authors · 2022-11-18
> **Response to Reviewer E5p2**
>
> > are there other complex bounds that can be used to learn analytic functions?
>
> Please see [the general response](https://openreview.net/forum?id=UcKEodTPtfI&noteId=ycRp1OZSni) for a discussion of our choice of this particular bound.
>
> > The frequency of these three paths is in the order: nighttime>daytime>twilight, but the complexity order is: nighttime < twilight < daytime. Could you explain how this happened?
>
> Frequency refers to prevalence in the underlying data distribution (and not in the optimal sampling distribution). In this case, the most frequent path (in the underlying data distribution) is the least complex, and the least frequent path is the most complex. Frequency in the underlying data distribution and path complexity are both used as inputs when calculating the optimal sampling distribution. We have clarified this point in Section 2 in the revision.
>
> > Page 4, Figure 2 (b). It seems that when the total data size is small (around 10^1) the error decreases faster under the frequency sampling. Could you provide a possible reason for this?
>
> At this small number of samples (total dataset size = 15), the training process is somewhat noisy (even with 10 trials), which can be seen in the nighttime path’s error increasing when increasing from 4 samples to 5 samples. This noise leads to our sampling approach resulting in slightly higher error at this data point.
>
> > Page 4, "Training methodology". Is the proposed method sensitive to these parameters?  How do these parameters influence the result?
>
> We tuned these hyperparameters to those that maximized accuracy across all paths. Training with less optimized hyperparameters would increase the error of both our method and of the baseline. Assuming that suboptimal hyperparameters independently increased the error of each path by a constant factor, the relative performance improvement of our method over the baseline would remain constant. A more fine-grained analysis of hyperparameter sensitivity would require a more detailed understanding of the impact of hyperparameters on neural network performance, which is an open problem outside the scope of this work.
>
> > Instead of stochastic gradient decent, can other methods be used?
>
> The complexity bounds from Agarwala (2021) assume wide 1-layer ReLU nets trained to convergence with SGD. In preliminary experiments we have observed that they empirically hold to the same extent with other optimizers and network architectures.
>
> >  Please provide the definition of "probably approximately correctly learnable".
>
> The definition is given in Equation 1 in the submission. We have edited the text in the revision to be more clear about this.
>
> > Compared with these existing languages, what are the advantages of the presented language?
>
> Please see [the general response](https://openreview.net/forum?id=UcKEodTPtfI&noteId=ycRp1OZSni) for a discussion of Turaco as a standalone programming language.
>
> > Could you please list the pros and cons of your offline method compared with the online method in Coetes et al (2019)? In addition, could you explain how the different choice on the sample complexity affects the result?
>
> These approaches do not so much have relative tradeoffs (pros and cons) as they do different underlying assumptions about the learning setting:
>
> The definition of complexity ($\zeta$ in our formalism) that Cortes et al. use is a function of the number of hypotheses in the hypothesis class, the total number of data points used, and the number of data points for a given stratum that have been queried thus far. It is not a function of any complexity metric of the function being learned.
>
> Cortes et al.'s approach is thus a better fit when learning stratified functions of unknown complexity (i.e., non-analytic functions) using a finitely sized hypothesis class (not a neural network, since neural networks induce an effectively infinitely large hypothesis class), and is targeted at the online setting assuming they are given a sampler of the overall data distribution but not one for each stratum. Our approach is a better fit when learning stratified analytic functions with neural networks, and is targeted at the offline setting assuming we are given a sampler for each stratum.
>
> We have included this more precise discussion in the revision.

---

### Official Review · Reviewer_mndd · 2022-11-02

**Confidence:** 4
**Correctness:** 1
**Technical Novelty And Significance:** 2
**Empirical Novelty And Significance:** 1
**Recommendation:** 1

**Clarity, Quality, Novelty And Reproducibility:**

The paper has inconsistencies, certain things are used before they are introduced (e.g. the language for the didactic example in figure 1). I do not see an easy way to attempt to reproduce the results presented in the paper.

**Strength And Weaknesses:**

Strengths: the paper takes a more thorough approach than earlier works to distribution of samples for training program surrogates. Theoretical error bounds are used to derive practical sample distribution rules. The results are empirically supported.

Weaknesses: the paper has many technical inconsistencies which impair significantly my trust in the presented results.

First, the phrase 'optimal sampling' is used through the paper, including formal statements, while it is clear that the sample distribution is based on theoretical error bounds, which are not tight, and has sampling is not 'optimal'.

Second, the proof, provided in the appendix, of the central theorem of the paper, 3.1, contains errors. 1) the derivative of the 'Lagrangian' is wrong, the derivative of n^{-1/2} is -1/2n^{-3/2} rather than 1/2n^{-3/2} (the minus is omitted). 2) the last line is obviously wrong. 3) this is not a proof of constrained optimization but rather a sketch of it, for a normal proof one has to show that the zero-gradient point is indeed the minimum (it does not have to be).

Third, the formula in Theorem 3.1 has a paradoxical property --- as the number c of strata/branches increases, their relative complexity affects 'optimal' sample distribution less and less. I am not convinced that this is indeed 'optimal' behavior rather than an artifact of a particular form of upper bound used for the proof (and more suitable for theoretical analysis than for practical sampling). This should at least be discussed.

Fourth, there is no supplementary material, so one cannot reproduce the results, or even look at the source code. The only artifact of the empirical evaluation is the 'full code' of the renderer case study in Figure 18, which is a single function written in a version of Turaco different from introduced in the paper.

**Summary Of The Paper:**

The paper proposes improved sampling for training program surrogates. The proposed sampling scheme takes into account both data distribution and sample complexity of different paths. A language suitable for application of the proposed sampling scheme is introduced, and the scheme is evaluated on a graphics program.

**Summary Of The Review:**

The paper addresses an interesting topic but does not stand by publication standards.

---

> ### Author Response · Authors · 2022-11-18
> **Response to Reviewer mndd**
>
> > the phrase 'optimal sampling' is used through the paper, including formal statements, while it is clear that the sample distribution is based on theoretical error bounds, which are not tight, and has sampling is not 'optimal'.
>
> The paper clearly states the objective that our sampling approach is optimal with respect to: "minimizing the upper bound on the stratified surrogate's error", in Section 1; “the upper bound of Equation (3) is minimized at:” in Theorem 3.1. To further make this clear, we have updated the first bullet point in the contributions section of the revision to state that our sampling approach minimizes an upper bound on the error.
>
> Optimizing upper/lower bounds is standard practice across multiple communities when calculating the objective itself is intractable, such as in variational inference [1], adversarial training [2, 3], and neural network generalization bounds [4]. In cases where the bound is insufficiently tight, there is a risk that an approach that is optimal with respect to the upper bound does not improve the ultimate objective. However, establishing exact sample complexity results is an open problem for deep neural networks.
>
> Please see [the general response](https://openreview.net/forum?id=UcKEodTPtfI&noteId=ycRp1OZSni) above for a discussion of our choice of this particular bound and for a discussion of the tightness of this upper bound.
>
> > the proof, provided in the appendix, of the central theorem of the paper, 3.1, contains errors. 1) the derivative of the 'Lagrangian' is wrong … 2) the last line is obviously wrong
>
> Thank you for identifying these typos in Appendix C; we have corrected the proof in the revision. Theorem 3.1 is still true and the main body of the paper remains unchanged.
>
> > 3) this is not a proof of constrained optimization… one has to show that the zero-gradient point is indeed the minimum
>
> The objective is convex, and therefore the local minimum is the global minimum. We have included this statement in the revision in Appendix C.
>
> > Third, the formula in Theorem 3.1 has a paradoxical property… I am not convinced that this is indeed 'optimal' behavior…
>
> With the stated assumption ($\delta_i = \frac{\delta}{c}$), Theorem 3.1 is optimal with respect to the upper bound in Equation 2. Please again see [the general response](https://openreview.net/forum?id=UcKEodTPtfI&noteId=ycRp1OZSni) above for a discussion of our choice of this particular bound.
>
> > Fourth, there is no supplementary material
>
> Please see [the general response](https://openreview.net/forum?id=UcKEodTPtfI&noteId=ycRp1OZSni) above for a link to the source code for the implementation and experiments.
>
> ### References
>
> [1] Lawrence K. Saul, Tommi Jaakkola, Michael I. Jordan. Mean Field Theory for Sigmoid Belief Networks. Journal of Artificial Intelligence Research, 4, 1996. https://arxiv.org/abs/cs/9603102
>
> [2] Eric Wong, J. Zico Kolter. Provable defenses against adversarial examples via the convex outer adversarial polytope. International Conference on Machine Learning, 2018. https://arxiv.org/abs/1711.00851
>
> [3] Aditi Raghunathan, Jacob Steinhardt, Percy Liang. Semidefinite relaxations for certifying robustness to adversarial examples. Conference on Neural Information Processing Systems, 2018. https://arxiv.org/abs/1811.01057
>
> [4] Gintare Karolina Dziugaite, Daniel M. Roy. Computing Nonvacuous Generalization Bounds for Deep (Stochastic) Neural Networks with Many More Parameters than Training Data. Conference on Uncertainty in Artificial Intelligence, 2016. https://arxiv.org/abs/1703.11008

---

> > ### Comment · Reviewer_mndd · 2022-11-20
> > **soundness and signficiance**
> >
> > Soundness: You mentioned optimizing the lower bound in variational inference. This is essentially different from your case. In variational inference, optimizing the lower bound is optimal in the sense that if you maximized the lower bound over a family of functions, then you found the optimal solution given the family. The lower bound is not a 'statistical' bound, it is a term in the sum equal to the optimization target, in which the other term is unknown but constant. In your case, the bound is not tight, or not shown to be tight. Therefore, optimizing according to your chosen bound may lead to the choice of a suboptimal solution. Thus, I still maintain that your claim of optimality is unsupported.
> >
> > Significance: a basic analysis of the bound equation shows that as c grows, the influence of relative complexity of different branches vanishes. Therefore,  optimization according to the proposed bound only makes  a difference for a small number of branches. In more complicated settings with many branches (of potentially varying complexity), sample distribution computed according to your method approaches sample distribution that ignores sample complexity of different branches. Therefore, I still maintain that the contribution of very limited significance.
> >
> > I can see how, in principle, an algorithm can be devised that accounts for sample complexity by allocating samples to different branches and improves inference performance due to that, but this is not the case with the algorithm you propose.

---

> > > ### Author Response · Authors · 2022-11-22
> > > **Response to Reviewer mndd**
> > >
> > > Thanks to the reviewer for the continued discussion.
> > >
> > > > ...variational inference...
> > >
> > > We apologize if we were unclear on this point. The reviewer correctly notes that in the textbook formulation of variational inference, optimizing the ELBO is equivalent to optimizing the KL divergence [1, Section 2.2]. We did not mean to imply otherwise. However, many approaches within the field of variational inference optimize bounds on the likelihood which do not directly optimize either the divergence or likelihood itself. This is the case in the paper cited in our original response [2; see Equation 19 and Figures 5 and 6] as well as in [3; see Section 4.2, Section 6.5, and Figure 2 (right)] and [4; see Equation 4].
> > >
> > > > ...your claim of optimality is unsupported.
> > >
> > > As stated in the response, the paper is clear on the scope of this claim ("minimizing the upper bound on the stratified surrogate's error", in Section 1; “the upper bound of Equation (3) is minimized at:” in Theorem 3.1). We would welcome suggestions on additional places to clarify this point.
> > >
> > > If the main concern is not the approach but the specific use of the word “optimal”, we would also be happy to remove the word “optimal'' from the title and to revise the paper to not use the word “optimal'' in isolation without making it clear that we are optimizing the upper bound.
> > >
> > > > In more complicated settings with many branches (of potentially varying complexity), sample distribution computed according to your method approaches sample distribution that ignores sample complexity of different branches.
> > >
> > > This is indeed a property of our bound: if the number of paths is significantly larger than the complexity of each path, our sampling distribution approaches the uniform distribution. This arises from the PAC form of the underlying sample complexity bound: in cases where any individual component of the stratified surrogate fails to train successfully, we cannot bound the error of the stratified surrogate. This property does not contradict any theoretical or empirical results presented in the paper. We will include this discussion in the final version of the paper.
> > >
> > > > I can see how, in principle, an algorithm can be devised that accounts for sample complexity by allocating samples to different branches and improves inference performance due to that, but this is not the case with the algorithm you propose.
> > >
> > > This is not true. Section 5 shows that the algorithm we propose, which accounts for sample complexity by allocating samples to different branches, improves inference performance.
> > >
> > > ## References
> > >
> > > [1] David M. Blei, Alp Kucukelbir, Jon D. McAuliffe. Variational Inference: A Review for Statisticians. Journal of the American Statistical Association, Vol. 112, 2017. https://arxiv.org/abs/1601.00670
> > >
> > > [2] Lawrence K. Saul, Tommi Jaakkola, Michael I. Jordan. Mean Field Theory for Sigmoid Belief Networks. Journal of Artificial Intelligence Research, 4, 1996. https://arxiv.org/abs/cs/9603102
> > >
> > > [3] Chris J. Maddison, Dieterich Lawson, George Tucker, Nicolas Heess, Mohammad Norouzi, Andriy Mnih, Arnaud Doucet, Yee Whye Teh. Filtering Variational Objectives. ICML, 2018. https://arxiv.org/abs/1705.09279
> > >
> > > [4] Rajesh Ranganath, Dustin Tran, David M. Blei. Hierarchical Variational Models. ICML, 2016. https://arxiv.org/pdf/1511.02386.pdf

---

> > > > ### Comment · Reviewer_mndd · 2022-11-24
> > > > **unconvinced**
> > > >
> > > > I am respectfully unconvinced by your arguments.
> > > >
> > > > In particular,
> > > >
> > > > 1. I am unconvinced that calling optimizing a loose upper bound "optimal sampling" is **sound**. In my opinion, this is misleading for the reader.
> > > >
> > > > 2. I am unconvinced that an algorithm that aims to improve over uniform sampling but converges to uniform sampling as the number of sampling sites grows is a **significant** contribution.

---

### Author Response · Authors · 2022-11-18
**Response to all reviewers**

Thanks to all reviewers for their feedback. We have updated the revision with changes requested by the reviewers. All changes are highlighted in yellow.

Below we respond to concerns shared among multiple reviewers.

## Motivation for the specific upper bound chosen (Reviewers mndd, E5p2):
> [Reviewer mndd:] the formula in Theorem 3.1 has a paradoxical property… I am not convinced that this is indeed 'optimal' behavior rather than an artifact of a particular form of upper bound used for the proof

> [Reviewer E5p2:] are there other complex bounds that can be used to learn analytic functions?

We chose the specific complexity bound used in Section 2 for two primary reasons. First, it is well established in the literature after being introduced by Arora et al. (2019). Second, the compositional nature of the bound (Equation 4) lends itself to calculation via program analysis. We are not aware of other complexity bounds that share these properties.

To the best of our knowledge, our paper is the first to calculate and use sample complexity bounds to improve training of surrogates of programs. While finding and evaluating alternative bounds would be interesting, it is outside of the scope of this work.

## Tightness of upper bounds (Reviewers mndd, E5p2, RLWc):
> [Reviewer mndd:] the sample distribution is based on theoretical error bounds, which are not tight…

> [Reviewer RLWc:] it would be good to know something about how tight these bounds are.

Agarwala et al. (2021) provide empirical and theoretical results on the tightness of the bounds in Appendices B.2 and C. We bolster these results in Figure 2(a) showing a correlation between the theoretically calculated complexity and empirically measured sample complexity. Our results in Section 5 broadly demonstrate that optimizing with respect to these upper bounds results in accuracy improvements.

## Turaco as a programming language (Reviewers mndd, E5p2, RLWc):

> [Reviewer mndd:] …written in a version of Turaco different from introduced in the paper.

> [Reviewer E5p2]: Compared with these existing languages, what are the advantages of the presented language?

> [Reviewer RLWc:] The Turaco programming language lacks iterative constructs like loops or recursion, which is very important for programming. Could they also be included, and how difficult would it be?

Turaco demonstrates the core techniques required to perform the complexity analysis. We therefore do not present constructs peripheral to the complexity analysis like the vector extension noted on page 7, or loops or recursion. Extending the implementation to such constructs is an engineering task requiring no additional research (with the caveat that loops and recursion can induce a potentially infinite number of paths in the program, so the technique would be limited to those programs with finitely many paths).

Rather than considering Turaco as a standalone programming language, a practical deployment of Turaco could embed the language into an existing language (like how PyTorch is embedded into Python). Performing such an embedding for Turaco is purely an engineering task requiring no additional research.

## Source code for implementation and evaluation (Reviewers mndd, RLWc):

> [Reviewer mndd]: there is no supplementary material

> [Reviewer RLwc:] the implementations of the Turaco programming language and the methodology to compute the complexity bounds of the traces should be made public.

We have released the source code of the Turaco interpreter and analysis, the neural network training code, and the renderer evaluation on Github at https://github.com/iclr-2022-UcKEodTPtfI/turaco. We have included a pointer to this repository in Section 4 of the revision.

## References

Atish Agarwala, Abhimanyu Das, Brendan Juba, Rina Panigrahy, Vatsal Sharan, Xin Wang, and Qiuyi Zhang. One network fits all? modular versus monolithic task formulations in neural networks. ICLR, 2021.

Sanjeev Arora, Simon Du, Wei Hu, Zhiyuan Li, and Ruosong Wang. Fine-grained analysis of optimization and generalization for overparameterized two-layer neural networks. ICML, 2019.

---

### Author Response · Authors · 2022-12-06
**Response to reviewer updates**

Thanks to all reviewers for continuing to evaluate the paper.

## Usage of the term "optimal"

> [Reviewer mndd:] calling optimizing a loose upper bound 'optimal sampling' ... is misleading for the reader

> [Reviewer E5p2:] the paper seems not very rigorous in defining "optimal sampling"...

> [Reviewer RLWc:]  calling the method "optimal sampling" ... [is a] significant issue

All reviewers have raised issues with the terminology in the paper. As mentioned in our response to Reviewer mndd, we would be happy to remove the word "optimal" from the title and to revise the paper to not use the word "optimal" in isolation without making it clear that we are optimizing the upper bound. For example, we would call our approach "complexity-guided sampling" rather than "optimal sampling". We have supplied the AC with a draft of the paper with this change applied. We additionally note that this change in wording does not change the strength or other implications of our results.

Please let us know if this does not address the concern about the term "optimal sampling", and if not then in what specific ways it does not.

## Significance of behavior in the limit

> [Reviewer mndd:] I am unconvinced that an algorithm that aims to improve over uniform sampling but converges to uniform sampling as the number of sampling sites grows is a significant contribution.

> [Reviewer E5p2:] ... the contributions of the paper seem not significant.

> [Reviewer RLWc:] ... the method converging to uniform sampling [is a] significant issue

We first note an error we did not catch in our previous response to Reviewer mndd: the approach does not converge to the uniform distribution, but instead to a distribution with each path being sampled proportionally to its frequency raised to the power of 2/3.

With that, we have included the discussion below verbatim in the updated draft supplied to the AC at the end of Section 3.2:

> In the limit of infinite strata ($\lim_{c \to \infty}$), the complexity-guided sampling approach induced by Theorem 3.1 converges to sampling each stratum with probability proportional to $D(s_i)^{\frac{2}{3}}$. In the limit, this distribution does not account for complexity. However in practice the complexity still guides sampling. Section 5 evaluates an example with all complexities $\zeta(f_i) \geq 5899$ and $\delta=0.01$. For the $\log(c \delta^{-1})$ term to match the contribution of the complexity term there would need to be $\approx 10^{2600}$ strata; this example only has 9. Thus while in the infinite limit of strata our approach is complexity-agnostic, in practice it is dominated by the complexity.

This analysis, along with our empirical results in Section 5 (about which reviewers have raised no concerns), validates the significance of our results in practice. Again, please let us know if this does not address concerns about the significance of the technique, and if not then in what specific ways.

---

### Decision · Program_Chairs · 2023-01-20

**Decision:**

Reject

**Justification For Why Not Higher Score:**

There were sufficient concerns from the reviewers that they felt the paper deserved a revision with a full new round of review.

**Justification For Why Not Lower Score:**

N/A

**Metareview: Summary, Strengths And Weaknesses:**

This paper was a borderline paper which led to quite extensive discussion among the reviewers (and was selected for a virtual meeting). The initial scores from the three reviewers were 8/8/1, obviously representing a fairly severe divergence. The negative review identified a few technical errors; these were largely addressed by the authors in their update and in the rebuttal. However, this kicked off a broader discussion among the reviewers regarding the relevance of the theory. Ultimately the consensus was that the revised paper represented a "major revision", which would be best served by a fresh round of review at the next appropriate conference. More details are in the summary of the AC-reviewer meeting below.

**Summary Of Ac-Reviewer Meeting:**

While some of the initial concerns were dealt with, there were a few open questions. The major concern was regarding the limiting behavior of the algorithm as the number of branches increases; if there are many paths, then the authors initially argued that this would approach a uniform distribution over paths; after some discussion this was revised to be the distribution corresponding to normalizing the path probabilities raised to 2/3. Beyond the revision in the theory reducing overall confidence, it wasn't entirely clear how quickly one would reach this limiting regime (where a natural baseline would be simple uncertainty sampling). The authors argued that this would be uncommon, but that is an empirical claim, not a theoretical one, and should be justified accordingly in experiments or in analysis of typical problems.

Two reviewers additionally expressed concern that the approach is not necessarily "optimal", in that the bound that is being optimized may or may not be tight.